# ΔNp63 regulates a common landscape of enhancer associated genes in non-small cell lung cancer

Marco Napoli [1,2,6], Sarah J. Wu[1,2,6], Bethanie L. Gore[1,2], Hussein A. Abbas [1,2], Kyubum Lee[3], Rahul Checker[1,2], Shilpa Dhar[4], Kimal Rajapakshe[5], Aik Choon Tan[3], Min Gyu Lee[4✉], Cristian Coarfa [5✉] & Elsa R. Flores [1,2✉]

Distinct lung stem cells give rise to lung adenocarcinoma (LUAD) and squamous cell carcinoma (LUSC). ΔNp63, the p53 family member and p63 isoform, guides the maturation of these stem cells through the regulation of their self-renewal and terminal differentiation; however, the underlying mechanistic role regulated by ΔNp63 in lung cancer development has remained elusive. By utilizing a ΔNp63-specific conditional knockout mouse model and xenograft models of LUAD and LUSC, we found that ΔNp63 promotes non-small cell lung cancer by maintaining the lung stem cells necessary for lung cancer cell initiation and progression in quiescence. ChIP-seq analysis of lung basal cells, alveolar type 2 (AT2) cells, and LUAD reveals robust ΔNp63 regulation of a common landscape of enhancers of cell identity genes. Importantly, one of these genes, *BCL9L*, is among the enhancer associated genes regulated by ΔNp63 in Kras-driven LUAD and mediates the oncogenic effects of ΔNp63 in both LUAD and LUSC. Accordingly, high *BCL9L* levels correlate with poor prognosis in LUAD patients. Taken together, our findings provide a unifying oncogenic role for ΔNp63 in both LUAD and LUSC through the regulation of a common landscape of enhancer associated genes.

[1] Department of Molecular Oncology, H. Lee Moffitt Cancer Center and Research Institute, Tampa, FL, USA. [2] Cancer Biology and Evolution Program, H. Lee Moffitt Cancer Center and Research Institute, Tampa, FL, USA. [3] Department of Biostatistics and Bioinformatics, H. Lee Moffitt Cancer Center and Research Institute, Tampa, FL, USA. [4] Department of Molecular and Cellular Oncology, University of Texas MD Anderson Cancer Center, Houston, TX, USA. [5] Department of Molecular and Cell Biology, Baylor College of Medicine, Houston, TX, USA. [6]These authors contributed equally: Marco Napoli, Sarah J. Wu. ✉email: mglee@mdanderson.org; coarfa@bcm.edu; elsa.flores@moffitt.org

Lung cancer is the number one cause of cancer mortality worldwide and non-small-cell lung cancer (NSCLC) comprises 85% of all lung cancers, with adenocarcinoma (LUAD) and squamous cell carcinoma (LUSC) being the most common subtypes. The poor prognosis and lack of effective treatment of lung cancer identify the need both to better understand the molecular mechanisms underlying its progression and to develop new therapeutic modalities. Due to similarities between tumour cells and stem cells in both signalling pathways and self-renewal abilities, cancer may originate from transformed stem cells[1]. This hypothesis has been applied to the progenitor cell populations of the lung, including basal cells, alveolar type 2 (AT2) cells and bronchioalveolar stem cells (BASCs), which are involved in the regeneration of distinct regions of the lung[2]. The basal cells of the trachea differentiate into ciliated and goblet cells[3] and are hypothesised to be the cell of origin for LUSC, since both populations express similar histological markers, such as p63 and keratin 5 (Krt5)[4]. AT2 cells and BASCs maintain the distal lung alveolar epithelium and can initiate LUAD[5,6].

The transcription factor p63 is a member of the p53 family expressed in the basal cells of the lung[7] and is a master regulator of stem cell maintenance and differentiation of all epithelial tissues[8–12]. The characterisation of the role of p63 in the lung epithelium is complex; and, prior to this study, had yet to be elucidated. There are two N-terminal isoforms, TAp63 and ΔNp63, with distinct regulatory roles in stem cell populations in the skin: TAp63 regulates self-renewal of skin-derived precursors (SKPs) in the dermal papilla[11], while ΔNp63 performs this function in epidermal basal cells[12]. Similarly, the p63 isoforms exert diverse functions in cancer. While TAp63 serves as a tumour and metastasis suppressor in multiple tumour types[13,14], we and others have shown that ΔNp63 functions as an oncogene by inhibiting the functions of p53, TAp63 and TAp73[8,15]. Importantly, we showed that p53 deficient tumours require ΔNp63 for their maintenance and we have also identified novel ways to therapeutically target p53 deficient and mutant cancers through manipulation of ΔNp63[15–17].

In lung cancer, ΔNp63 is used in the diagnosis of LUSC[18] and was identified as a significantly altered gene in 44% of primary LUSCs by the Cancer Genome Atlas (TCGA)[19]. ΔNp63 has been demonstrated to be oncogenic in cutaneous SCC, which shares molecular characteristics and p63 expression patterns with LUSC[20]. The role of p63 in the distal lung has not been characterised to date because of its low expression; however, p63+/Krt5+ distal airway stem cells have been reported to regenerate distal lung after damage with influenza virus[21], thus indicating an important role for p63 in the distal lung. Furthermore, we found that the ΔNp63 transcriptional signatures generated from either keratinocytes or LUSC and LUAD identified ΔNp63 as an oncogene in both NSCLC subtypes[13]. Taken together, there is a critical need to understand the contribution of ΔNp63 in stem cells of the lung and how it may impact the initiation and progression of non-small cell lung cancer.

Using xenograft models of LUSC as well as the Kras-driven model of LUAD (Kras^{LSL-G12D/+})[22] intercrossed with mice carrying conditionally deleted ΔNp63, we demonstrate that ΔNp63 is required for the maintenance of progenitors of NSCLC. By analysing the role of ΔNp63 in AT2 and basal cells, respective cells of origin for these tumour types, we found that ΔNp63 is required for the self-renewal and maintenance of these cells and does so through the regulation of a common landscape of enhancer-associated genes. This landscape is also maintained in Kras-driven LUAD and includes BCL9L, a known oncogene[23–25] with expression levels that are prognostic in LUAD patients. Taken together, our findings unveil a crucial role for ΔNp63 in driving a common oncogenic transcriptional programme essential for the formation and progression of both LUAD and LUSC, and identify BCL9L as an important mediator of the oncogenic effects of ΔNp63 in both NSCLC subtypes.

## Results

**ΔNp63 promotes tumour initiation and progression in non-small cell lung cancer.** While the contribution of ΔNp63 in lung development has been described in refs. [26,27], its role in lung tumorigenesis is not well understood. In particular, little is known about the role of ΔNp63 in LUAD, primarily due to its low expression in this tumour type[28]. To investigate the functions of ΔNp63 in LUAD, we crossed the Kras-driven LUAD mouse model (Kras^{LSL-G12D/+})[22] to the ΔNp63 conditional knockout mouse model (ΔNp63^{fl/fl})[12]. The recombination was induced through the administration of intratracheal adenoviral Cre-recombinase[29], which led to a robust recombination efficiency in the trachea and lung parenchyma assessed by a switch from tdTomato (red) to GFP (green) expressed by the ROSA allele[30]. Twenty weeks postinfection with adenoviral Cre, we assessed a greater than 90% recombination in the trachea and distal lung assessed by GFP expression from the ROSA allele (Supplementary Fig. 1a). We found that Kras^{G12D/+} mice formed lung adenomas and a few LUAD as previously reported[22] (Fig. 1a). In contrast, lungs from ΔNp63^{Δ/Δ};Kras^{G12D/+} mice had almost no scorable adenomas or LUAD formation (Fig. 1a). Examination and grading of the lung lesions[22] revealed a twofold reduction in atypical adenomatous hyperplastic (AAH) lesions, and a fivefold reduction in grade 1 and grade 2+ in ΔNp63^{Δ/Δ};Kras^{G12D/+} mice compared to Kras^{G12D/+} mice (Fig. 1b). Tumours from Kras^{G12D/+} mice stained positively yet heterogeneously for ΔNp63 (Fig. 1c and Supplementary Fig. 1b, i), indicating that ΔNp63 is indeed expressed in Kras-driven LUAD.

Given the well-established role of ΔNp63 in stem cell maintenance in various epithelial tissues[9,11,21,31,32], we examined the role of ΔNp63 in the maintenance of lung stem cells. LUAD has been shown to originate from distal lung stem cells that express SPC or both CCSP and SPC[5,6]. Therefore, we performed immunofluorescence in adenomas and adenocarcinomas derived from Kras^{G12D/+} and ΔNp63^{Δ/Δ};Kras^{G12D/+} mice using the distal lung stem cell markers SPC and CCSP. Notably, we found that ΔNp63^{Δ/Δ};Kras^{G12D/+} tumours had a 50% reduction in CCSP+/SPC+ cells compared to tumours from Kras^{G12D/+} mice, suggesting that ΔNp63 serves to maintain the proliferation of distal lung stem cell populations (Fig. 1d). To assess whether ΔNp63 plays a similar oncogenic role in human LUAD, we performed soft agar assays in three LUAD cell lines expressing Kras^{G12D} (H1944, H358 and H2009) to assess anchorage-independent growth as a surrogate for transformation. Downregulation of ΔNp63 decreased colony formation ability in these three LUAD cell lines compared to the respective control cells (Fig. 1e and Supplementary Fig. 1j). Altogether, these data indicate that LUAD depends on ΔNp63 for their formation and progression.

We next evaluated the oncogenic functions of ΔNp63 in the context of lung squamous cell carcinoma (LUSC) by knocking down ΔNp63 in two human LUSC cell lines, H520 and H2170, and performing soft agar assays. Knockdown of ΔNp63 decreased the ability of H520 and H2170 cell lines to form colonies by 50% compared to colonies from the respective control cell lines (Fig. 1f, g and Supplementary Fig. 1k). Consistent with these in vitro results, knockdown of ΔNp63 resulted in a two to fivefold reduction in the in vivo tumour formation of the H520 and H2170 cell lines in xenograft mouse models (Fig. 1h, i). Histological analysis of tumours from xenografts indicated that

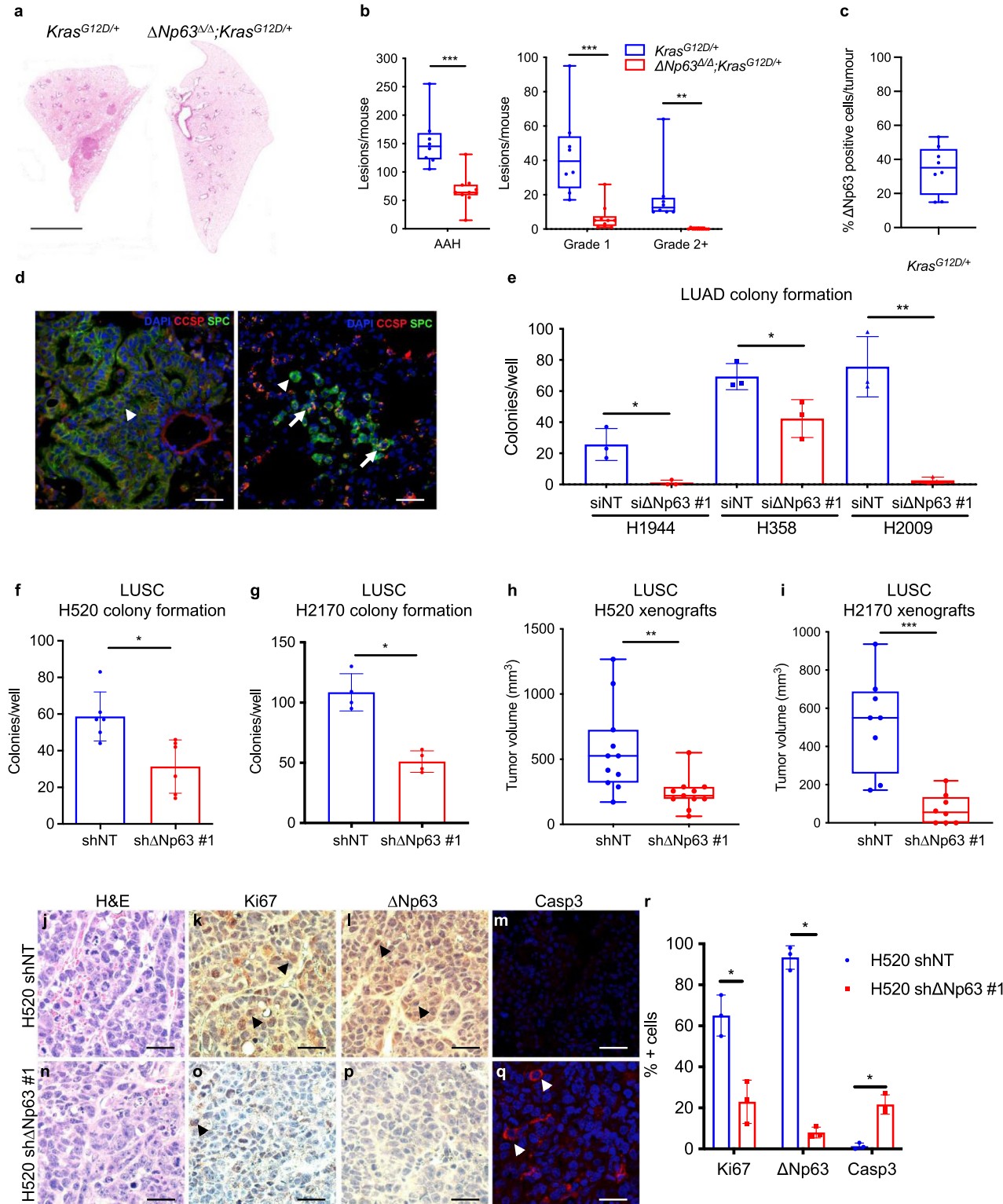

downregulation of *ΔNp63* caused a threefold decrease in proliferation, as assessed by Ki67, and a 20-fold increase in apoptosis, as assessed by cleaved caspase 3, compared to LUSC tumours expressing *ΔNp63* (Fig. 1j–r). Our laboratory has recently identified a keratinocyte-derived signature of *ΔNp63*-regulated genes and an oncogenic *ΔNp63*-driven transcriptional signature in the context of LUSC and LUAD[13]. These *ΔNp63* controlled transcriptional programmes support the role of *ΔNp63* as an oncogene in LUSC, as *ΔNp63* is highly amplified in LUSC[19]

and has been utilised as a positive diagnostic marker for LUSC[18]. Our data with the H520 and H2170 cell lines and xenograft models further support the oncogenic role of *ΔNp63* in the context of LUSC, indicating that LUSC cells depend on *ΔNp63* for their anchorage-independent growth and xenograft LUSC formation.

Taken together, our data provide evidence that *ΔNp63* is required for the formation and maintenance of both LUSC and LUAD.

**Fig. 1 ΔNp63 promotes tumour initiation and progression in non-small cell lung cancer. a** Representative H&E stained cross-sections of lung lobes from $Kras^{G12D/+}$ and $\Delta Np63^{\Delta/\Delta};Kras^{G12D/+}$ mice. Scale bar equals 3 mm. **b** Quantification of the tumour grading of $Kras^{G12D/+}$ and $\Delta Np63^{\Delta/\Delta};Kras^{G12D/+}$ LUAD. AAH atypical adenomatous hyperplasia. LUAD are divided into grade 1 and grade 2+. $n = 8$ mice, ** vs. $Kras^{G12D/+}$, $P < 0.001$, *** vs. $Kras^{G12D/+}$, $P < 0.0001$, two-tailed Student's $t$-test. **c** Representative image of IHC for ΔNp63 in lung lesions from $Kras^{G12D/+}$ mice. Scale bar equals 100 μm. **d** Representative staining for CCSP and SPC in LUAD from $Kras^{G12D/+}$ and $\Delta Np63^{\Delta/\Delta};Kras^{G12D/+}$ mice. Arrow for a single marker, arrowhead for a double marker. Scale bar equals 100 μm. **e** Quantification of colony formation efficiency in soft agar of the indicated LUAD cell lines transfected with the indicated siRNAs. Data are mean ± SD, $n = 3$, * vs. siNT, $P < 0.05$, ** vs. siNT, $P < 0.001$, two-tailed Student's $t$-test. **f, g** Quantification of colony formation efficiency in soft agar of the indicated LUSC cell lines infected with the indicated shRNAs. Data are mean ± SD, $n = 6$ (**f**) and $n = 4$ (**g**), * vs. shNT, $P < 0.05$, two-tailed Student's $t$-test. **h, i** Tumour volume quantification of xenograft tumours derived from the same cells as in (**f, g**). $n = 11$ (**h**) and $n = 8$ (**i**), ** vs. shNT, $P < 0.001$, *** vs. shNT, $P < 0.0001$, two-tailed Student's $t$-test. **j–q** Representative images of the xenograft tumours described in **h** showing H&E stained cross-sections (**n**), and staining for Ki67 (**k and o**), ΔNp63 (**l and p**), and cleaved caspase 3 (**m and q**). Scale bar equals 100 μm. **r** Quantification of the stainings shown in **k–m** and **o–q**. Data were mean ± SD, $n = 11$, * vs. shNT, $P < 0.05$, two-tailed Student's $t$-test. All boxplots represent the individual data points, median and whiskers (min to max method). Source data are provided as a Source Data file.

**In vivo ΔNp63 ablation in the tracheal epithelium results in acute increased proliferation leading to long-term depletion of Krt5+ basal cells.** ΔNp63 is highly expressed in the basal cells of the trachea, a cell type shown to be the cell of origin for LUSC[33]. To determine whether ΔNp63 serves to maintain LUSC through the regulation of tracheal basal stem cells, we utilised $\Delta Np63^{fl/fl}$ mice crossed with the Rosa reporter mouse[30] to enable identification of cells that have undergone cre-recombination. To specifically ablate ΔNp63 in tracheal and lung epithelium, we performed an intratracheal instillation of $\Delta Np63^{fl/fl};Rosa^{M/M}$ mice with adenoviral cre-recombinase with adenoviral empty vector serving as a negative control. Histological analysis of the trachea and lung tissues of the mice was performed at two timepoints, 1 month and 3 months postinfection. At 1-month postinfection, quantification of H&E stained sections of $\Delta Np63^{\Delta/\Delta};Rosa^{\Delta/\Delta}$ tracheal epithelium showed a fivefold increase of basal cells within the stratified epithelium (SE) and a fourfold increase of epithelial separation (ES) compared to the $\Delta Np63^{fl/fl};Rosa^{M/M}$ control (Fig. 2a, b). Staining for the basal marker keratin 5 (Krt5) revealed that the composition of the stratified epithelium in $\Delta Np63^{\Delta/\Delta};Rosa^{\Delta/\Delta}$ tracheal epithelium primarily contained Krt5+ basal cells, whereas the $\Delta Np63^{fl/fl};Rosa^{M/M}$ control maintained a single basal layer of Krt5+ cells (Fig. 2c, d). Krt5+ cells in $\Delta Np63^{\Delta/\Delta};Rosa^{\Delta/\Delta}$ tracheas were also fivefold more proliferative as measured by Ki67 staining compared to control (Fig. 2e, f). Intriguingly, this increased proliferation was associated with a strong increase in apoptosis. Indeed, up to 60% of cells were cleaved caspase 3 positive in the $\Delta Np63^{\Delta/\Delta};Rosa^{\Delta/\Delta}$ mice compared to almost no cleaved caspase 3 staining in their wild-type counterparts ($\Delta Np63^{fl/fl};Rosa^{M/M}$) (Fig. 2g, h), suggesting that these cells may undergo depletion or exhaustion over time. Therefore, we assessed the trachea and lung tissues at 3 months postinfection. At that timepoint, the majority of the $\Delta Np63^{\Delta/\Delta};Rosa^{\Delta/\Delta}$ tracheal epithelium exhibited a hypoplastic phenotype (HE) compared to the pseudostratified columnar appearance of the $\Delta Np63^{fl/fl};Rosa^{M/M}$ epithelium (Fig. 2i, j), indicating significant depletion in tracheal epithelial cells lacking ΔNp63. Krt5 staining of $\Delta Np63^{\Delta/\Delta};Rosa^{\Delta/\Delta}$ tracheal epithelium showed little to no Krt5+ basal cells (Fig. 2k, l), further indicating a depletion or exhaustion of basal cells lacking ΔNp63. To determine whether the loss of basal cells occurred due to apoptosis, we stained the epithelium for the apoptotic marker, cleaved caspase 3, which showed that the increase in apoptosis observed at the 1-month timepoint was also observed at the 3-month timepoint, when up to 10% of cells were cleaved caspase 3 positive (Fig. 2o, p). The defects in the tracheal epithelium and the staining for Krt5, Ki67 and cleaved caspase 3 were quantified in at least three mice per genotype both at the 1-month (Fig. 2q, r) and 3-month timepoints (Fig. 2s, t). Taken together, our data show that ΔNp63 is critical to maintain tracheal basal cells in quiescence and that loss of ΔNp63 leads to hyperproliferation followed by exhaustion of tracheal basal epithelial cells.

**ΔNp63 is required for the regeneration and terminal differentiation of the tracheal epithelium.** ΔNp63 has been shown to be a crucial regulator of terminal differentiation in the skin[11]. To determine whether ΔNp63 is similarly required for regeneration and terminal differentiation of tracheal basal cells, we tested if ΔNp63-depleted basal cells are able to differentiate into ciliated and goblet cells in vivo. To achieve this goal, we treated $\Delta Np63^{fl/fl};Rosa^{M/M}$ mice with either adenoviral cre or adenoviral empty. After 3 days, mice were administered polidocanol (PDO)[34] to deplete suprabasal tracheal epithelial cells or PBS as a control (Fig. 3a). We then analysed the regeneration of the tracheal epithelium at two timepoints after the PDO treatment, 7 days and 30 days. While the wild-type ($\Delta Np63^{fl/fl};Rosa^{M/M}$) basal cells were able to differentiate into a pseudostratified columnar structure of the epithelium including goblet (muc5ac positive) and ciliated (acetylated tubulin positive) cells within 7 days after the PDO challenge, the $\Delta Np63^{\Delta/\Delta};Rosa^{\Delta/\Delta}$ tracheal epithelial cells instead formed a disorganised epithelium of Krt5+ cells and lacked expression of muc5ac and acetylated tubulin (Fig. 3b–i). Notably, these Krt5+ cells in the $\Delta Np63^{\Delta/\Delta};Rosa^{\Delta/\Delta}$ tracheal epithelium were highly proliferative with elevated apoptosis compared to the wild-type ($\Delta Np63^{fl/fl};Rosa^{M/M}$) control trachea (Fig. 3j–m). These data indicate that the proliferation of $\Delta Np63^{\Delta/\Delta};Rosa^{\Delta/\Delta}$ tracheal epithelium at the 1-month timepoint post adenoviral cre injection (see Fig. 2f) is primarily due to basal cell proliferation and not compensatory dedifferentiation of the ciliated or goblet cells.

To determine the long-term consequences of ΔNp63 loss in tracheal epithelial cells, we examined the tracheal epithelium at 30 days after PDO administration and found that the $\Delta Np63^{\Delta/\Delta};Rosa^{\Delta/\Delta}$ mice treated with PDO had a thin, hypoplastic epithelium lacking Krt5+ cells (Fig. 3n–q). By performing immunostaining for terminal differentiation markers of tracheal epithelial cells, we found that $\Delta Np63^{\Delta/\Delta};Rosa^{\Delta/\Delta}$ tracheal epithelial cells were positive for acetylated tubulin similarly to the wild-type ($\Delta Np63^{fl/fl};Rosa^{M/M}$) control tracheas (Fig. 3r, s), but were devoid of goblet cells (Fig. 3t, u). Additionally, in contrast to the 7-day timepoint, at 30 days post-PDO administration, there were few Ki67 positive and cleaved caspase 3 positive in both genotypes (Fig. 3v–y). The staining for Krt5, acetylated tubulin, muc5ac, Ki67 and cleaved caspase 3 were quantified in at least three mice per genotype both at the 7-day (Fig. 3z) and 30-day timepoints (Fig. 3a′). Taken together, these data demonstrate the crucial role of ΔNp63 in the maintenance of tracheal basal cells in a quiescent state and the regulation of their terminal differentiation into a goblet and ciliated cells.

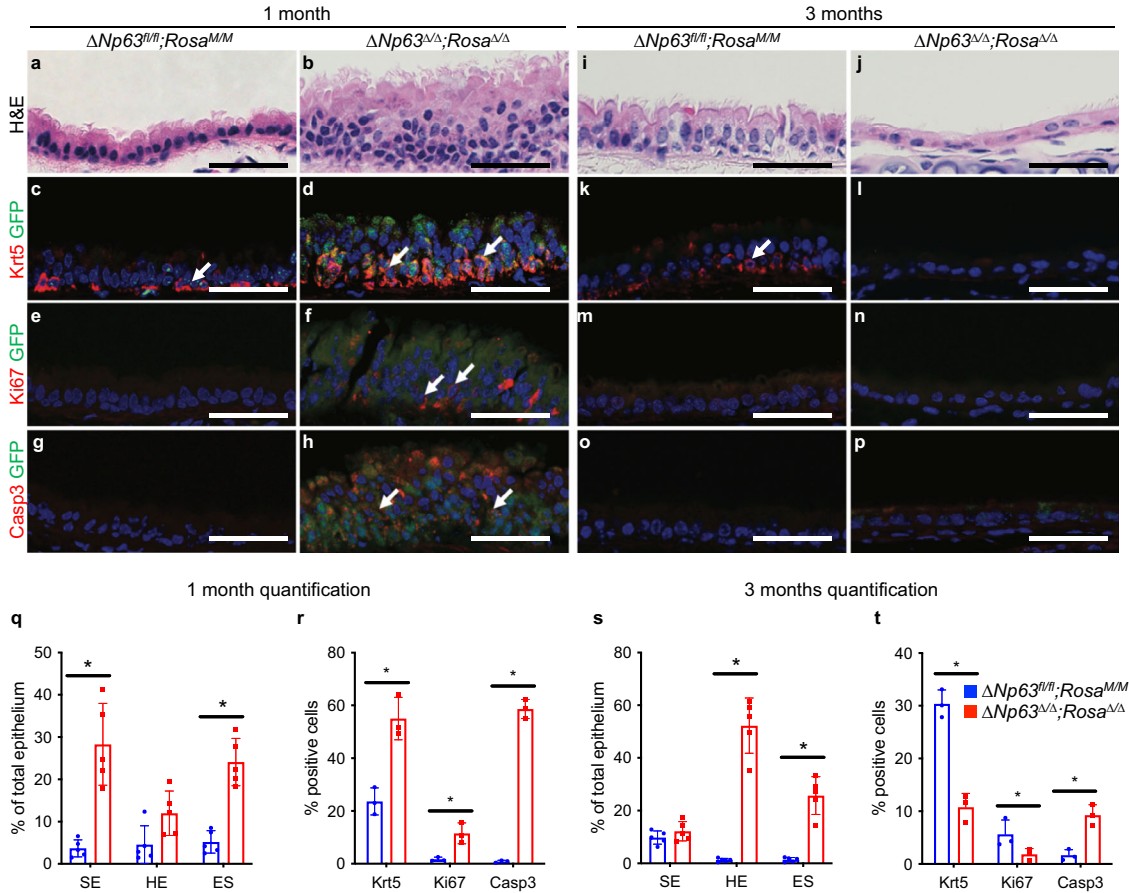

**Fig. 2 In vivo ΔNp63 ablation in the tracheal epithelium results in acute increased proliferation leading to long-term depletion of Krt5+ basal cells. a, b** Representative H&E stained cross-sections of ΔNp63$^{fl/fl}$;Rosa$^{M/M}$ and ΔNp63$^{Δ/Δ}$;Rosa$^{Δ/Δ}$ tracheal epithelia at 1-month postinfection. **c–h** Representative staining for Krt5/GFP (**c, d**), Ki67/GFP (**e, f**), and Casp3/GFP (**g, h**), in the same tracheal epithelia shown in **a, b. i, j** Representative H&E stained cross-sections of ΔNp63$^{fl/fl}$;Rosa$^{M/M}$ and ΔNp63$^{Δ/Δ}$;Rosa$^{Δ/Δ}$ tracheal epithelia at 3 months postinfection. **k–p** Representative staining for Krt5/GFP (**k, l**), Ki67/GFP (**m, n**), and Casp3/GFP (**o, p**), in the same tracheal epithelia shown in (**i, j**). All scale bars equal 100 μm. **q** Quantification of the stratified epithelium (SE), the hypoplastic epithelium (HE), and epithelial separation (ES) in ΔNp63$^{fl/fl}$;Rosa$^{M/M}$ and ΔNp63$^{Δ/Δ}$;Rosa$^{Δ/Δ}$ tracheal epithelia at 1-month postinfection. Data were mean ± SD, $n = 5$, * vs. ΔNp63$^{fl/fl}$;Rosa$^{M/M}$, $P < 0.05$, two-tailed Student's $t$-test. **r** Quantification of the staining for Krt5, Ki67, and Casp3 for ΔNp63$^{fl/fl}$;Rosa$^{M/M}$ and ΔNp63$^{Δ/Δ}$;Rosa$^{Δ/Δ}$ tracheal epithelia at 1-month postinfection. Data were mean ± SD, $n = 3$, * vs. ΔNp63$^{fl/fl}$;Rosa$^{M/M}$, $P < 0.05$, two-tailed Student's $t$-test. **s** Quantification of the stratified epithelium (SE), the hypoplastic epithelium (HE), and epithelial separation (ES) in ΔNp63$^{fl/fl}$;Rosa$^{M/M}$ and ΔNp63$^{Δ/Δ}$;Rosa$^{Δ/Δ}$ tracheal epithelia at 3 months postinfection. Data were mean ± SD, $n = 5$, * vs. ΔNp63$^{fl/fl}$;Rosa$^{M/M}$, $P < 0.05$, two-tailed Student's $t$-test. **t** Quantification of the staining for Krt5, Ki67, and Casp3 for ΔNp63$^{fl/fl}$;Rosa$^{M/M}$ and ΔNp63$^{Δ/Δ}$;Rosa$^{Δ/Δ}$ tracheal epithelia at 3 months postinfection. Data were mean ± SD, $n = 3$, * vs. ΔNp63$^{fl/fl}$;Rosa$^{M/M}$, $P < 0.05$, two-tailed Student's $t$-test. Source data are provided as a Source Data file.

**ΔNp63 is required for self-renewal of basal and distal lung stem cells.** To further investigate the mechanism of ΔNp63 in basal stem cell maintenance, we first isolated basal cells from ΔNp63$^{fl/fl}$;Rosa$^{M/M}$ mouse tracheas and infected them with either adenoviral cre to deplete ΔNp63 or adenoviral empty as a control. Next, we cultured them in vitro on J2-3T3 feeder cells and incubated with 5-ethynyl-2'-deoxyuridine (EdU) to determine their proliferation capacity. While at passage 1, ΔNp63$^{Δ/Δ}$;Rosa$^{Δ/Δ}$ basal cell colonies replicated similarly to wild-type basal cells (Fig. 4a), at passage 2, they became hyperproliferative and exhibited a 5- to 10-fold increase in the >95%-positive EdU incorporation fraction compared to its wild-type counterparts (Fig. 4b). However, at passage 3, ΔNp63$^{Δ/Δ}$;Rosa$^{Δ/Δ}$ mouse tracheal cells were half as proliferative compared to the wild-type basal cells (Fig. 4c), thus mimicking the behaviour observed in vivo, where ΔNp63$^{Δ/Δ}$; Rosa$^{Δ/Δ}$ mouse tracheal cells are initially hyperproliferative and are then exhausted through apoptosis. Indeed, even in vitro, ΔNp63$^{Δ/Δ}$;Rosa$^{Δ/Δ}$ mouse tracheal cells are characterised by an increase in apoptosis compared to wild-type controls (Fig. 4d).

To investigate the effect of ΔNp63 loss on self-renewal, we cultured ΔNp63$^{fl/fl}$;Rosa$^{M/M}$ and ΔNp63$^{Δ/Δ}$;Rosa$^{Δ/Δ}$ mouse tracheal basal cells in 3D and assessed their efficiency in forming tracheospheres. The cells devoid of ΔNp63 formed significantly fewer spheres over time, which were also fivefold smaller in size than the tracheospheres expressing ΔNp63 (Fig. 4e–g). These effects were amplified by the passaging of the basal tracheospheres, indicating that ΔNp63 is required for the self-renewal of tracheal basal stem cells.

To determine whether ΔNp63 is required for tracheal basal cell differentiation in vitro, we differentiated tracheal basal cells in 3D matrigel culture for 20 days[3]. While the wild-type tracheal basal cells formed hollow bi-layered spheres (Fig. 4h, i) comprised of an exterior layer of Krt5+ and NGFR+ basal cells (Fig. 4j, k) and an interior of ciliated and goblet cells (Fig. 4l, m), ΔNp63$^{Δ/Δ}$;Rosa$^{Δ/Δ}$ tracheal basal cells instead formed 100-fold smaller, mono-layered spheres (Fig. 4n, o) that lacked staining for differentiation markers, including Krt5, acetylated tubulin and muc5ac (Fig. 4p–s). While the ΔNp63$^{Δ/Δ}$;Rosa$^{Δ/Δ}$ tracheospheres lacked

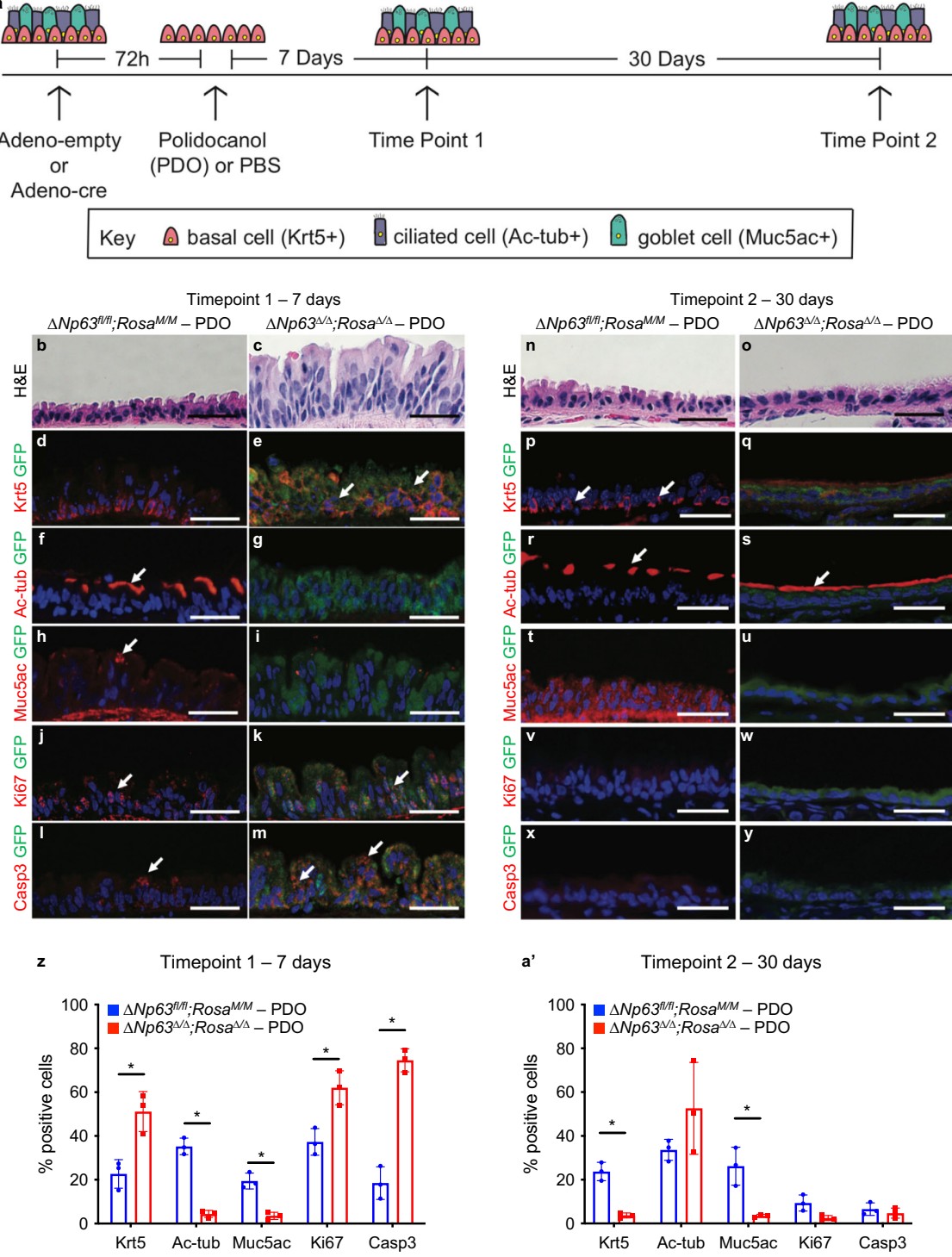

**Fig. 3 ΔNp63 is required for the regeneration and terminal differentiation of the tracheal epithelium. a** Schematic of polidocanol tracheal injury experiment. **b, c** Representative H&E stained cross-sections of $\Delta Np63^{fl/fl};Rosa^{M/M}$ and $\Delta Np63^{\Delta/\Delta};Rosa^{\Delta/\Delta}$ tracheal epithelia at 7 days post-PDO treatment. **d–m** Representative staining for Krt5/GFP (**d, e**), Ac-tub/GFP (**f, g**), Muc5ac/GFP (**h, i**), Ki67/GFP (**j, k**) and Casp3/GFP (**l, m**) in the same tracheal epithelia shown in **b, c. n, o** Representative H&E stained cross-sections of $\Delta Np63^{fl/fl};Rosa^{M/M}$ and $\Delta Np63^{\Delta/\Delta};Rosa^{\Delta/\Delta}$ tracheal epithelia at 30 days post-PDO treatment. **p–y** Representative staining for Krt5/GFP (**p, q**), Ac-tub/GFP (**r, s**), Muc5ac/GFP (**t, u**), Ki67/GFP (**v, w**) and Casp3/GFP (**x, y**) in the same tracheal epithelia shown in (**n, o**). All scale bars equal 100 μm. **z** Quantification of the staining for Krt5, Ac-tub, Muc5ac, Ki67 and Casp3 for $\Delta Np63^{fl/fl};Rosa^{M/M}$ and $\Delta Np63^{\Delta/\Delta};Rosa^{\Delta/\Delta}$ tracheal epithelia at 7 days post-PDO treatment. Data were mean ± SD, $n = 3$, * vs. $\Delta Np63^{fl/fl};Rosa^{M/M}$, $P < 0.05$, two-tailed Student's $t$-test. **a'** Quantification of the staining for Krt5, Ac-tub, Muc5ac, Ki67 and Casp3 for $\Delta Np63^{fl/fl};Rosa^{M/M}$ and $\Delta Np63^{\Delta/\Delta};Rosa^{\Delta/\Delta}$ tracheal epithelia at 30 days post-PDO treatment. Data were mean ± SD, $n = 3$, * vs. $\Delta Np63^{fl/fl};Rosa^{M/M}$, $P < 0.05$, two-tailed Student's $t$-test. Source data are provided as a Source Data file.

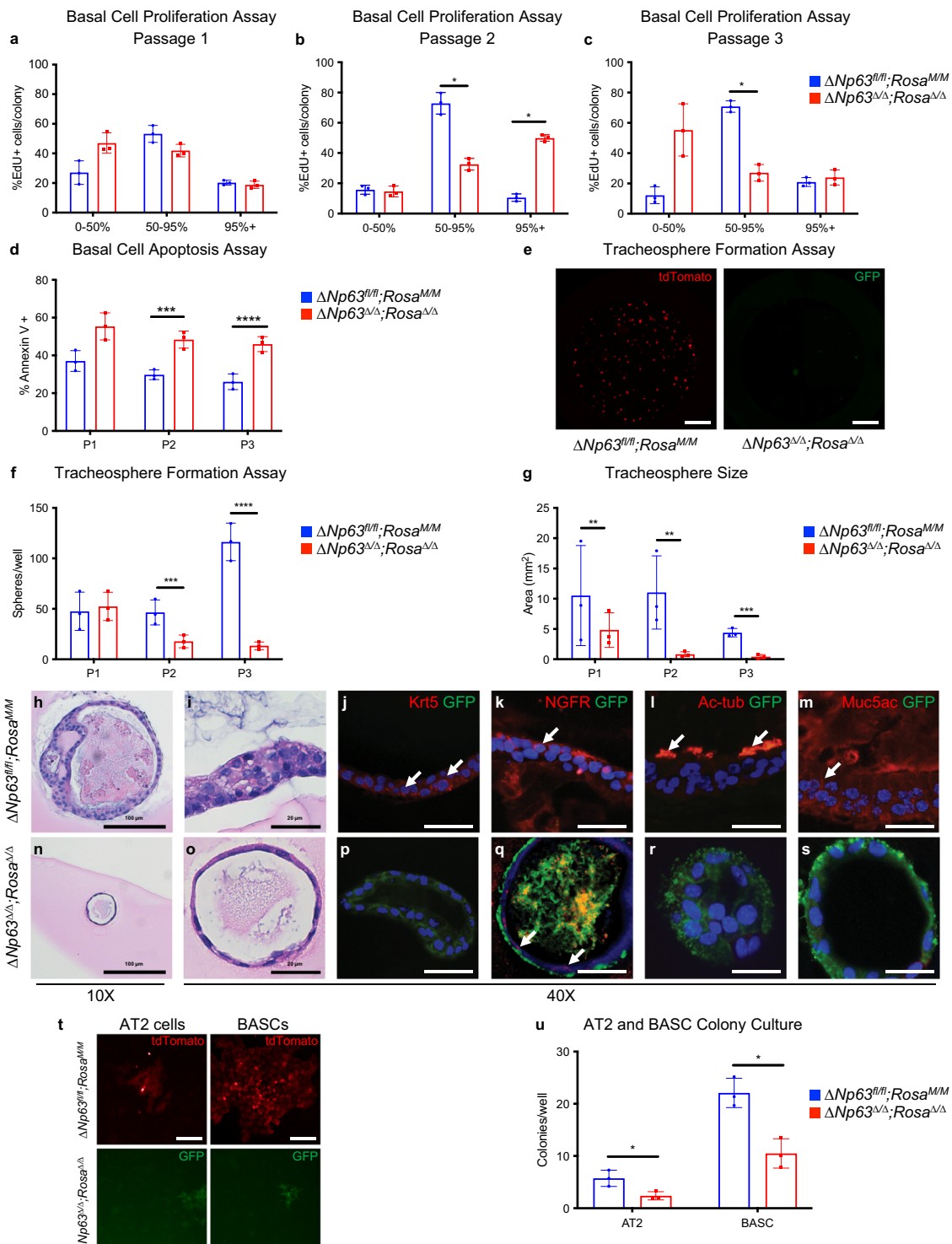

**Fig. 4 ΔNp63 is required for self-renewal of basal and distal lung stem cells. a–c** Quantification of EdU incorporation in $\Delta Np63^{fl/fl};Rosa^{M/M}$ and $\Delta Np63^{\Delta/\Delta};Rosa^{\Delta/\Delta}$ colonies at the indicated passages. **d** Quantification of Annexin V positive cells the same $\Delta Np63^{fl/fl};Rosa^{M/M}$ and $\Delta Np63^{\Delta/\Delta};Rosa^{\Delta/\Delta}$ colonies shown in (**a–d**). Data were mean ± SD, $n = 3$, * vs. $\Delta Np63^{fl/fl};Rosa^{M/M}$, $P < 0.05$, two-tailed Student's $t$-test. **e** Representative wells of tracheosphere formation assay of $\Delta Np63^{fl/fl};Rosa^{M/M}$ and $\Delta Np63^{\Delta/\Delta};Rosa^{\Delta/\Delta}$ basal cells at passage 3. Scale bars equal 250 μm. **f, g** Quantification of number (**f**) and size (**g**) of tracheospheres formed by $\Delta Np63^{fl/fl};Rosa^{M/M}$ and $\Delta Np63^{\Delta/\Delta};Rosa^{\Delta/\Delta}$ basal cells at the indicated passages. Data were mean ± SD, $n = 3$, * vs. $\Delta Np63^{fl/fl};Rosa^{M/M}$, $P < 0.05$, two-tailed Student's $t$-test. **h–s** Representative images of tracheospheres in differentiation assay generated from $\Delta Np63^{fl/fl};Rosa^{M/M}$ and $\Delta Np63^{\Delta/\Delta};Rosa^{\Delta/\Delta}$ basal cells, showing H&E stained cross-sections (**h, i** and **n, o**), and staining for Krt5/GFP (**j** and **p**), NGFR/GFP (**k** and **q**), Ac-tub/GFP (**l** and **r**) and Muc5ac/GFP (**m** and **s**). **t** Representative images of colonies formed by $\Delta Np63^{fl/fl};Rosa^{M/M}$ and $\Delta Np63^{\Delta/\Delta};Rosa^{\Delta/\Delta}$ AT2 cells and BASCs. Scale bars equal 50 μm. **u** Quantification of the same colonies as in (**t**). Data were mean ± SD, $n = 3$, * vs. $\Delta Np63^{fl/fl};Rosa^{M/M}$, $P < 0.05$, two-tailed Student's $t$-test. Source data are provided as a Source Data file.

staining for the basal cell marker, Krt5 (Fig. 4p), an alternative marker for basal cells NGFR was expressed (Fig. 4q), demonstrating that cells lacking *ΔNp63* originate from a Krt5 negative basal lineage. Taken together, our data indicate that *ΔNp63* is required for self-renewal and terminal differentiation of tracheal basal cells.

Because we found that *ΔNp63* is required for the formation and progression of LUAD and LUSC and has a key role in maintaining tracheal basal cells, we next asked whether *ΔNp63* may also regulate distal lung stem cells, where the cell of origin of LUAD reside. Therefore, we isolated AT2 and BASC cells from wild-type (*ΔNp63^{fl/fl};Rosa^{M/M}*) lungs and infected them with adenoviral empty or adenoviral cre to delete *ΔNp63*. By performing a colony formation assay using *ΔNp63^{fl/fl};Rosa^{M/M}* and *ΔNp63^{Δ/Δ};Rosa^{Δ/Δ}* AT2 and BASC cells, we found that the loss of *ΔNp63* resulted in a twofold decrease in colony formation of AT2 cells and BASCs (Fig. 4t, u). Taken together, our results demonstrate that *ΔNp63* is essential for the proper maintenance of the different lung stem cell populations, including basal cell, AT2, and BASC cells, and for the terminal differentiation of tracheal basal cells into a goblet and ciliated cells.

**ΔNp63 regulates the enhancer landscape of cell identity genes in basal cells of the trachea.** Many transcription factors required for the self-renewal, maintenance and identity of progenitor and stem cells have been shown to regulate the enhancer landscape of genes that control these processes[35–37]. These enhancers, marked by acetylated H3K27 (H3K27ac) in the presence of monomethyl H3K4 (H3K4me1), highly upregulate gene expression and are linked to the regulation of cell identity, stem cell pluripotency and cancer states[37–39]. To verify whether *ΔNp63* regulates self-renewal and terminal differentiation of tracheal basal cells by controlling the enhancer landscape, we isolated primary tracheal basal cells from *ΔNp63^{fl/fl};Rosa^{M/M}* and *ΔNp63^{Δ/Δ};Rosa^{Δ/Δ}* and performed RNA-seq and chromatin immunoprecipitation (ChIP)-seq analysis (Fig. 5a). Pathway analysis was performed using the differentially expressed genes identified from the RNA-seq data. Most of the pathways affected in *ΔNp63* deficient tracheal basal cells are involved in proliferation, stem cell maintenance and adhesion (Fig. 5b and Supplementary dataset 1). To identify direct transcriptional targets of *ΔNp63* in the tracheal basal cells, we performed low-cell number ChIP-seq analysis using an antibody directed against *ΔNp63*, which allowed us to generate a *ΔNp63* binding motif similar to the previously published p63 motif[40] (Fig. 5c). To identify enhancers regulated by *ΔNp63*, we also performed ChIP-seq for RNA polymerase 2 (Pol2) and H3K27ac. Importantly, we found that lack of *ΔNp63* affected genome-wide H3K27ac and Pol2 occupancy, especially at the most active enhancers (Fig. 5d, e, Supplementary Fig. 2a and Supplementary dataset 2). Comparison of the H3K27ac ChIP-seq signal revealed that the top 500 enhancers were significantly decreased upon deletion of *ΔNp63*, with a higher H3K27ac signal observed at regions containing the *ΔNp63* motif (Fig. 5d). Among the genes associated with the top 2000 enhancers in basal cells, there are genes involved in epithelialization and cell junction maintenance (Fig. 5e, f and Supplementary dataset 2). In cells expressing *ΔNp63*, *ΔNp63*-regulated genes such as *Krt5* and *BCL9L* are associated with a robust H3K27ac signal, including in enhancer regions reported in the Fantom 5 enhancers[41] and the Mouse Encode lung enhancers[42] catalogues (Fig. 5g, h and Supplementary dataset 3). These data indicate that *ΔNp63* regulates the expression of lung basal cell identity genes by binding to and affecting the acetylation status of the corresponding enhancers.

**ΔNp63 regulates self-renewal by controlling the enhancer landscape of cell identity genes in distal lung stem cells.** Given the crucial role of *ΔNp63* in the regulation of the enhancer landscape of cell identity genes of tracheal basal cells, we asked whether *ΔNp63* is playing a similar role in lung distal progenitor cells. To address this, we performed low-cell number ChIP-seq for H3K27ac, Pol2 and *ΔNp63*, in AT2 cells isolated from *ΔNp63^{fl/fl};Rosa^{M/M}* lungs and infected with adenoviral empty or adenoviral cre to delete *ΔNp63* (Fig. 6a). Similar to what was found in basal cells (see Fig. 5c), also in AT2 cells the *ΔNp63* binding motif was comparable to the previously published p63 motif[40] (Fig. 6b). While we did not observe a genome-wide reduction in the H3K27ac occupancy (Supplementary Fig. 2b and Supplementary dataset 3), ranking of genes based on the H3K27ac signal identified genes associated with the top 2000 enhancers in AT2 cells that had decreased expression after the loss of *ΔNp63*, including *BCL9L* and *ETV5* (Fig. 6c, d and Supplementary dataset 4). Interestingly, *BCL9L* was controlled by *ΔNp63* also in the tracheal basal cells (see Fig. 5f), while *ETV5* is an essential gene for maintenance of the AT2 cell population[43]. Importantly, the ChIP-seq peaks revealed that the recruitment of *ΔNp63* is associated with regions of increased H3K27ac signal in both the *ETV5* and *BCL9L* loci (Fig. 6d, e). Taken together, our findings demonstrate that *ΔNp63* also regulates the enhancer landscape of cell identity genes in the distal lung stem cell population and that this regulation may be important for its oncogenic role.

**ΔNp63 regulates a common landscape of enhancer-associated genes in tracheal basal cells, AT2 cells and LUAD.** Given that *ΔNp63* regulates the enhancer landscape in tracheal basal cells and AT2 cells, we asked whether there were any common enhancers regulated in both cell types that may shed light on its function as a common regulator of stem cell identity and oncogenesis. Cross comparison of the datasets derived from tracheal basal cells and AT2 cells indicated that the top 2000 enhancer signatures from both cell types shared 5326 common genes (Fig. 7a and Supplementary dataset 5) that may mediate the common function of *ΔNp63* in tumour maintenance. To enable translation from our mouse data into human samples, we performed a mouse-human cross-species analysis of the identified genes using H3K27ac ChIP-seq data from human LUAD and LUSC. In each of these datasets, we ranked the top 2000 enhancers and queried genes located near the enhancer regions that also contained an *ΔNp63* binding motif. Genes that met these criteria were substantiated with the LUAD[44] and LUSC[19] TCGA databases for alterations in cancer. Using these criteria, we found *Krt5* (a basal cell-specific gene), *ETV5* (an AT2-specific gene) and *BCL9L* (one of the genes in common between the basal cell and the AT2 signatures) as significant in both mouse and human datasets.

To determine whether *ΔNp63* directly regulates the enhancers of cell identity genes, we employed a CRISPR-dCas9 (the nuclease-null deactivated Cas9) based system[45]. Specifically, we transfected 293 T cells with vectors expressing dCas9-conjugated *ΔNp63* that were targeted to enhancer regions via short guide RNAs (sgRNAs) corresponding to the *ΔNp63* motif (Fig. 7b). dCas9-p300 was also included, since the histone acetyltransferase p300 also marks the most active enhancer regions[46]. Four sgRNAs were designed for targeting an *ΔNp63* motif within the identified enhancer regions of the *Krt5*, *ETV5* and *BCL9L* loci. While there was no significant effect with dCas9-p300 alone at identified enhancers, dCas9-*ΔNp63* resulted in robust gene expression of *ETV5, BCL9L* and *KRT5* by 3, 7 and 13 folds

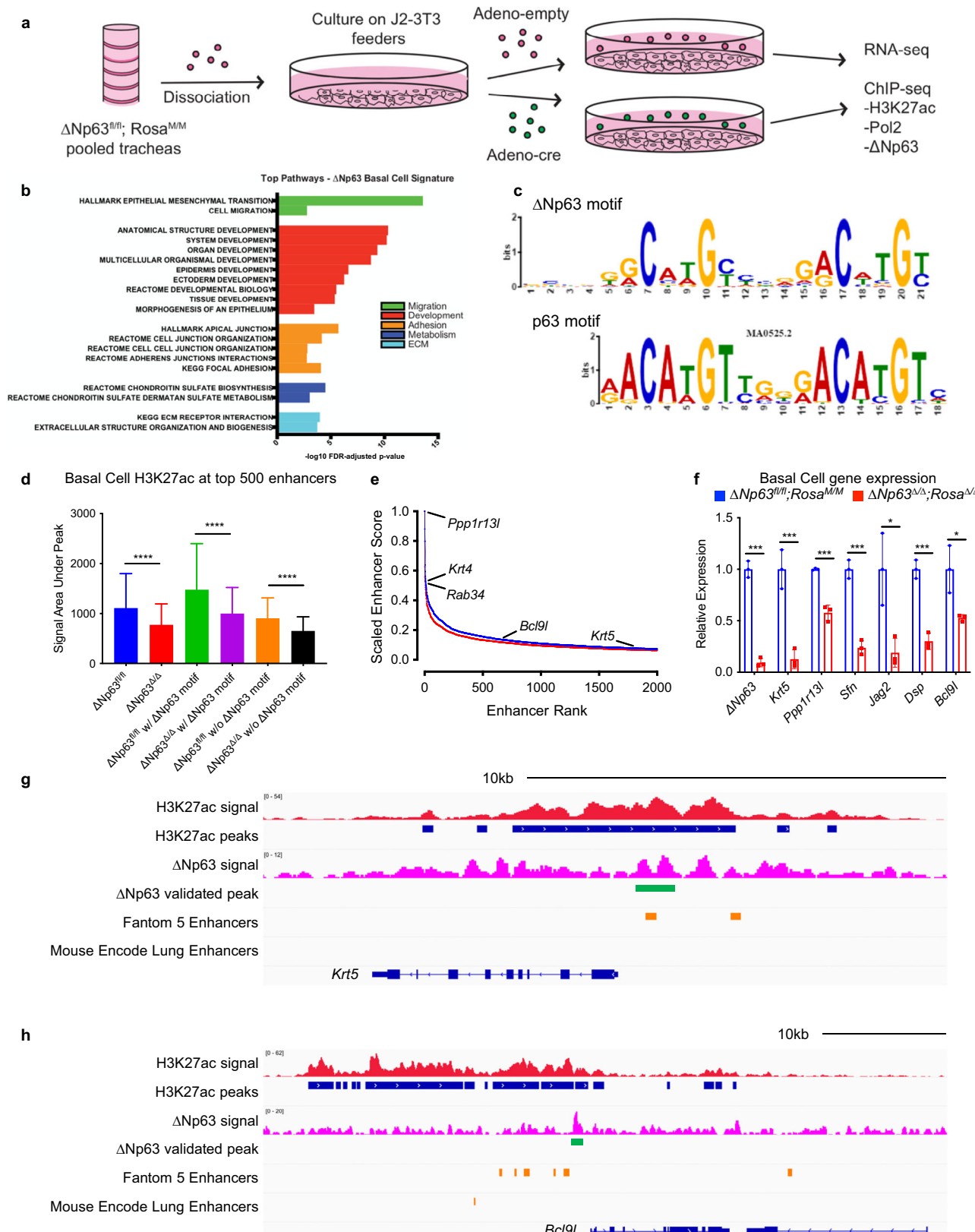

respectively (Fig. 7c–e). Moreover, the combination of dCas9-ΔNp63 and dCas9-p300 recruitment further increased gene expression significantly over dCas9-ΔNp63 alone, demonstrating that both ΔNp63 and p300 are recruited to these enhancer regions and are necessary to induce the expression of the corresponding gene (Fig. 7c–e).

To further evaluate whether ΔNp63 and p300 localise at ΔNp63 motifs located in enhancer regions, we performed ChIP in the H520 LUSC cell line, which expresses high levels of ΔNp63. When ΔNp63 was targeted via shRNA, the occupancy levels of H3K27ac were significantly decreased compared to the control at the enhancers of *ETV5* (4.5-fold), *BCL9L* (3.8-fold) and *KRT5*

**Fig. 5 ΔNp63 regulates the enhancer landscape of cell identity genes in basal cells of the trachea. a** Schematic of isolation of *ΔNp63^{fl/fl};Rosa^{M/M}* and *ΔNp63^{Δ/Δ};Rosa^{Δ/Δ}* basal cells for RNA-seq and ChIP-seq. **b** Pathway analysis of differentially expressed genes between *ΔNp63^{fl/fl};Rosa^{M/M}* and *ΔNp63^{Δ/Δ}; Rosa^{Δ/Δ}* basal cells. **c** ΔNp63 motif derived from basal cell ΔNp63 ChIP-seq compared to published p63 motif. **d** H3K27ac ChIP-seq signal area under peak for top 500 enhancers in the indicated samples. Data were mean ± SD, n = 3, **** vs. respective *ΔNp63^{fl/fl};Rosa^{M/M}*, P < 10^{−6}, two-tailed Student's t-test. **e** Ranking of top 2000 enhancers based on H3K27ac ChIP-seq in *ΔNp63^{fl/fl};Rosa^{M/M}* (blue) and *ΔNp63^{Δ/Δ};Rosa^{Δ/Δ}* (red) basal cells. **f** qRT-PCR of the indicated genes in *ΔNp63^{fl/fl};Rosa^{M/M}* and *ΔNp63^{Δ/Δ};Rosa^{Δ/Δ}* basal cells. Data were mean ± SD, n = 3, * vs. *ΔNp63^{fl/fl};Rosa^{M/M}*, P < 0.05, *** vs. *ΔNp63^{fl/fl}; Rosa^{M/M}*, P < 0.0001, two-tailed Student's t-test. **g, h** ChIP-seq profiles of *ΔNp63^{fl/fl};Rosa^{M/M}* basal cells for the indicated tracks in the *Krt5* (**g**) and *Bcl9l* (**h**) loci. Source data are provided as a Source Data file.

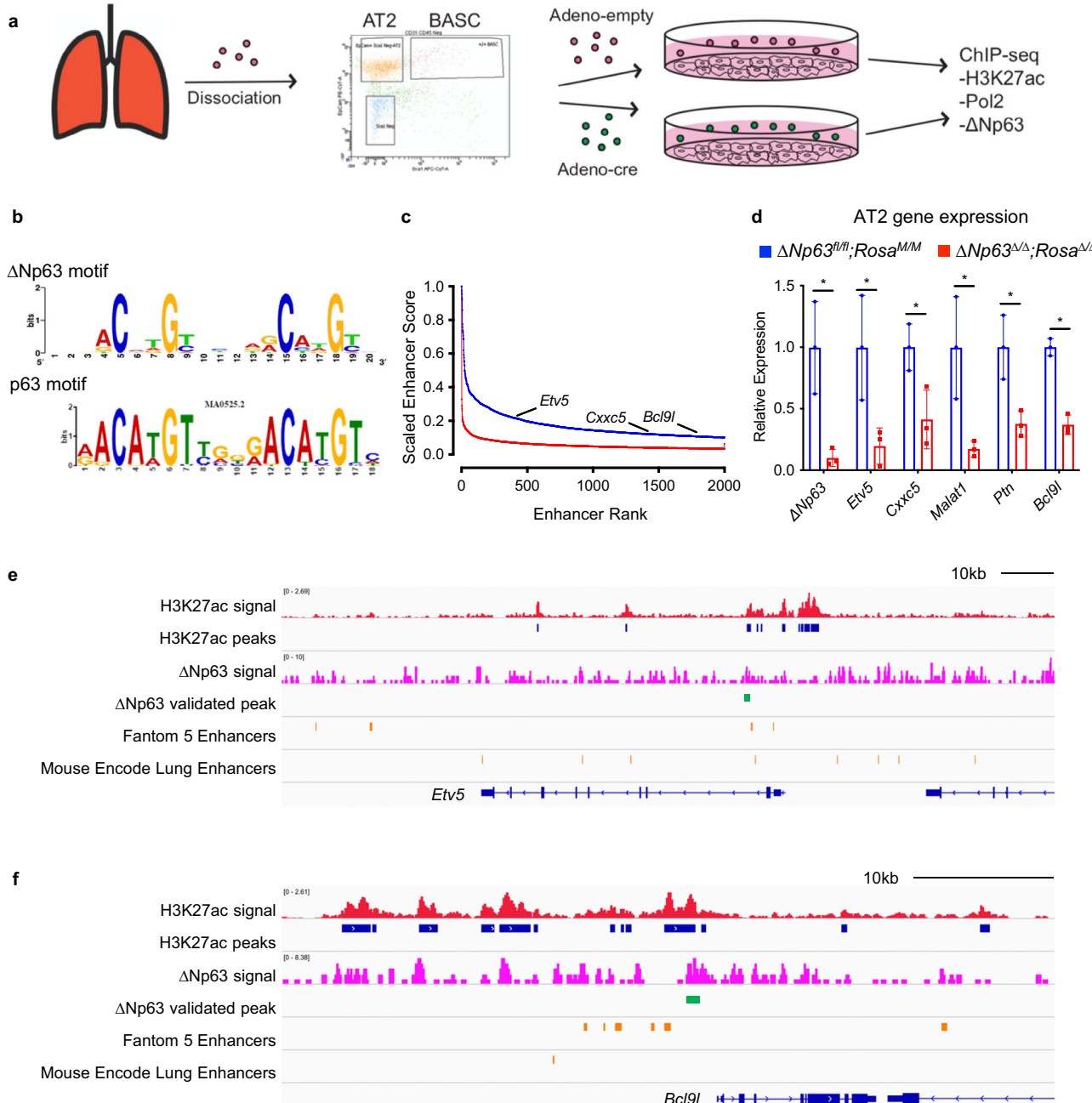

**Fig. 6 ΔNp63 regulates self-renewal by controlling the enhancer landscape of cell identity genes in distal lung stem cells. a** Schematic of isolation of *ΔNp63^{fl/fl};Rosa^{M/M}* and *ΔNp63^{Δ/Δ};Rosa^{Δ/Δ}* AT2 cells for RNA-seq and ChIP-seq. **b** ΔNp63 motif derived from AT2 cell ΔNp63 ChIP-seq compared to published p63 motif. **c** Ranking of top 2000 enhancers based on H3K27ac ChIP-seq in *ΔNp63^{fl/fl};Rosa^{M/M}* (blue) and *ΔNp63^{Δ/Δ};Rosa^{Δ/Δ}* (red) AT2 cells. **d** qRT-PCR of the indicated genes in *ΔNp63^{fl/fl};Rosa^{M/M}* and *ΔNp63^{Δ/Δ};Rosa^{Δ/Δ}* AT2 cells. Data were mean ± SD, n = 3, * vs. ΔNp63^{fl/fl}, P < 0.05, two-tailed Student's t-test. **e, f** ChIP-seq profiles of *ΔNp63^{fl/fl};Rosa^{M/M}* AT2 cells for the indicated tracks in the *Etv5* (**e**) and *Bcl9l* (**f**) loci. Source data are provided as a Source Data file.

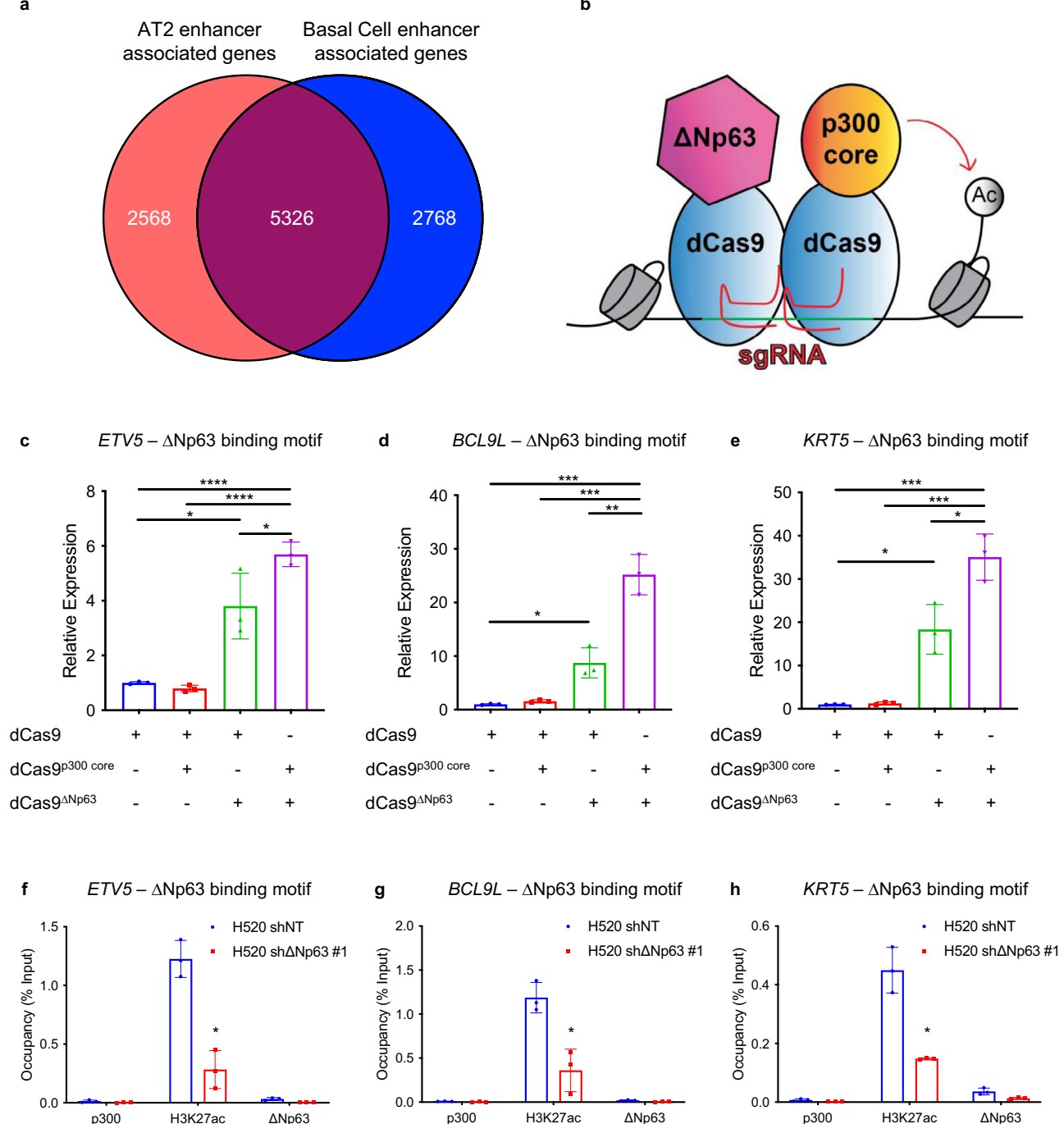

**Fig. 7 ΔNp63 regulates a common set of enhancer-associated genes in tracheal basal cells, AT2 cells and LUAD. a** Venn diagram overlaying the genes associated with the top 2000 enhancers in AT2 and basal cells. **b** Schematic of dCas9-p300 and dCas9-ΔNp63 sgRNA experiment. **c–e** Relative expression levels of *ETV5* (**c**), *BCL9L* (**d**) and *KRT5* (**e**) in 293 T cells over-expressing the indicated constructs. Data were mean ± SD, $n = 3$, *$P < 0.05$, **$P < 0.01$, ***$P < 0.001$, ****$P < 0.0001$, two-tailed Student's *t*-test. **f–h** qRT-PCR of ChIP assays for p300, H3K27ac and ΔNp63 at the indicated sites. Data were mean ± SD, $n = 3$, * vs. shNT, $P < 0.05$, two-tailed Student's *t*-test. Source data are provided as a Source Data file.

(2.8-fold) (Fig. 7f–h). Taken together, these results indicate that ΔNp63 cooperates with p300 to regulate the enhancer landscape of lung cell identity genes involved in LUAD and LUSC.

**BCL9L mediates the oncogenic activities of ΔNp63 and is critical for the maintenance of LUSC and LUAD.** The fact that ΔNp63 regulates common enhancer-associated genes in LUSC and LUAD suggests a common mechanism for its function as an

oncogene. To understand whether these genes are essential for the oncogenic functions of ΔNp63 in LUSC and LUAD, we used shRNA to knockdown expression of the basal cell-specific gene *KRT5*, the AT2-specific gene *ETV5* and *BCL9L*, which is one of the genes in common between the basal cell and the AT2 signatures, in the H520 LUSC cell line and the H358 LUAD cell line. Notably, in an in vitro soft agar assay, we found that knocking down the common gene, *BCL9L*, or the basal cell gene, *Krt5*, significantly reduced soft agar colony formation in the H520

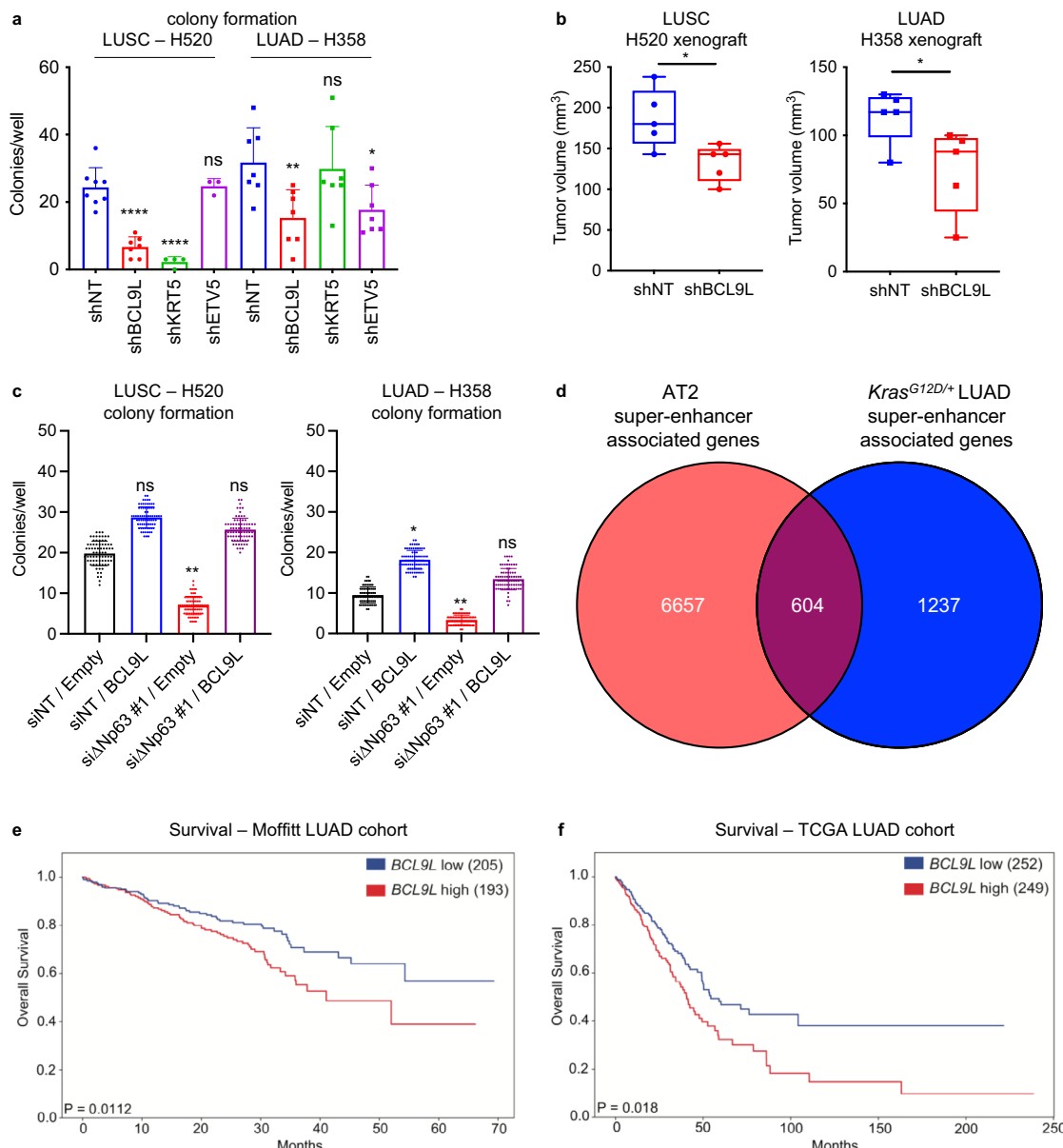

**Fig. 8 *BCL9L* mediates the oncogenic activities of ΔNp63 and is critical for maintenance of LUSC and LUAD. a** Quantification of colony formation efficiency in soft agar of H520 LUSC cells and H358 LUAD cells infected with the indicated shRNAs. Data were mean ± SD, *n* = 3, *P < 0.05, **P < 0.01, ****P < 0.0001, vs. respective shNT, two-tailed Student's *t*-test. **b** Tumour volume quantification of xenograft tumours derived from the indicated cell lines. Boxplots represent the individual data points, median and whiskers (min to max method) are shown, *n* = 5, * vs. respective shNT, P < 0.05, two-tailed Student's *t*-test. **c** Quantification of colony formation efficiency in soft agar of H520 LUSC cells and H358 LUAD cells transfected with the indicated constructs. Boxplots represent the individual data points, median and whiskers (min to max method) are shown, *n* = 3, *P < 0.05, **P < 0.01, vs. respective siNT/empty, two-tailed Student's *t*-test. **d** Venn diagram overlaying genes associated with the top 2000 enhancers in AT2 cells and LUAD derived from *Kras^{G12D/+}* mice. **e**, **f** Kaplan–Meier curves of overall lung adenocarcinoma survival data based on the levels of *BCL9L* in the indicated cohorts. Source data are provided as a Source Data file.

LUSC cell line, whereas knockdown of the AT2-specific gene, *ETV,5* did not produce a significant effect (Fig. 8a), thus indicating that regulation of *BCL9L* and *KRT5* by ΔNp63 is critical for maintenance of LUSC. Conversely, knocking down the common gene, *BCL9L*, or AT2-specific gene, *ETV5*, significantly reduced soft agar colony formation in the H358 LUAD cell line, whereas knockdown of the basal cell-specific gene, *KRT5*, did not produce a significant effect (Fig. 8a). These data indicate that *BCL9L* and *ETV5* are ΔNp63 targets critical for the maintenance of LUAD. To further demonstrate the importance of *BCL9L* in LUSC and LUAD in vivo, we injected the H520 LUSC cells or the H358 LUAD cells expressing shBCL9L or shNT used as a

negative control into the flank of nude mice. Importantly, we found that knockdown of *BCL9L* impaired tumour formation in both cell types (Fig. 8b), further indicating that *BCL9L* is an ΔNp63-regulated oncogene. To test the role of BCL9L in the pro-oncogenic activities of ΔNp63, we performed soft agar assays in two LUSC cell lines (H520 and H2170) and three LUAD cell lines (H358, H1944 and H2009). In all the tested cell lines, down-regulation of ΔNp63 decreased the expression levels of *BCL9L* (Supplementary Fig. 3a–f) and significantly reduced the in vitro colony formation (Fig. 8c and Supplementary Fig. 4a–e). Notably, the overexpression of myc-tag BCL9L was able to rescue the anchorage-independent ability in all the five cell lines (Fig. 8c and

Supplementary Fig. 4a–e), indicating that BCL9L is an important mediator of the oncogenic effects of ΔNp63 in both LUAD and LUSC. The oncogenic role of BCL9L in lung cancers is also supported by the comparison of ΔNp63-regulated enhancers associated genes in AT2 cells and in LUAD derived from tumours of $Kras^{G12D/+}$ mice, showing that $BCL9L$ is included in the common gene set (Fig. 8d and Supplementary dataset 6). Finally, to evaluate the relevance of $BCL9L$ as an oncogene in human lung cancers, we analysed its prognostic value in two independent cohorts of LUAD and LUSC. Importantly, in both LUAD cohorts, high levels of $BCL9L$ correlated with a poorer prognosis compared to low levels of $BCL9L$ (Fig. 8d, e). In both LUSC cohorts, a similar trend was observed, even though it was not significant. Taken together, this further indicates that the regulation of the $BCL9L$ enhancer landscape by ΔNp63 is critical for its function as an oncogene in both LUSC and LUAD and for the first time provides a common mechanism for the ΔNp63 functions as an oncogene in NSCLC.

## Discussion

Recent work has identified epigenetic mechanisms of gene regulation as key modulators of cancer progression[47]. In particular, enhancers are considered key regulators of spatiotemporal gene expression, with the most active enhancers being frequently associated with genes that designate cell identity[38]. Interest has grown in understanding the regulation of genes associated with the most active enhancers, since these genes are the most highly expressed in the cell and are thought to play a key role in the biology of cancer. Here, we demonstrate the crucial role of the transcription factor ΔNp63 in controlling the enhancer landscape of genes important for the maintenance of lung stem cell populations. Through this regulation, ΔNp63 exerts its oncogenic functions in both lung squamous cell carcinoma (LUSC) and adenocarcinoma (LUAD). Importantly, our findings demonstrate that ΔNp63 regulates a set of common genes, including $BCL9L$, through recruitment of ΔNp63 along with histone acetyl-transferases p300 to enhancer regions, in both LUSC and LUAD.

Our data are in line with previous reports showing that ΔNp63 is required for the maintenance of basal cells in the epidermis[12], where ΔNp63 regulates enhancer-associated genes essential for the homoeostasis[48] and differentiation[49] of the epidermis. Notably, the regulation of the enhancer landscape by ΔNp63 has also been found in epithelial tumours, including pancreatic adenocarcinomas, where the ΔNp63 controlled enhancer-associated genes drive squamous trans-differentiation[50]. Given the commonalities across squamous cell carcinomas of different tissues including LUSC[51], it is not surprising that analysis of LUSCs by the TCGA showed that ΔNp63 is overexpressed in almost half of all these primary tumours[19]. Conversely, until now, the involvement of ΔNp63 in the distal lung mainly relied on the findings that transiently amplified distal lung stem cells, which arise after lung injury, are positive for p63[21,52]. We now demonstrate a novel mechanism for ΔNp63 in AT2 cells, consisting of the upregulation of AT2 cell-specific genes through the ΔNp63-specific modulation of their enhancer landscape. Our LUAD mouse model obtained by combining the deletion of ΔNp63 with the $Kras^{G12D/+}$ mutation allowed us to demonstrate the tumour initiating role of ΔNp63, which is achieved through the regulation of stem cell signalling pathways that are often co-opted in cancer. Indeed, our findings suggest that ΔNp63 plays a critical role in the initiation and progression of cancers that are derived from stem cells requiring ΔNp63, including the lung basal cells and AT2 cells for LUSC and LUAD, respectively.

By performing ChIP-seq experiments in mouse lung stem cells and non-small cell lung tumours and then validating our findings

in human systems, including the TCGA LUSC[19] and LUAD[44] datasets, lung cancer xenografts, as well as an alternative exogenous model utilising dCas9 technology[53], we identified a group of enhancer-associated genes regulated by ΔNp63 in concert with p300. Among the genes controlled in both LUSC and LUAD, we found $BCL9L$, a member of the Wnt pathway[54] with known oncogenic activities in multiple cancer types, including colon adenocarcinomas[23], hepatocellular carcinomas[24] and pancreatic adenocarcinomas[25]. We now show that $BCL9L$ is regulated by ΔNp63 and acts as a crucial unifying oncogene in non-small cell lung cancers as well. Indeed, not only do high levels of $BCL9L$ correlate with poor overall survival in two independent cohorts of LUAD, but the overexpression of BCL9L can also rescue the reduced oncogenic growth caused by the downregulation of ΔNp63, thus indicating that BCL9L is an important common mediator of the oncogenic activities of ΔNp63 in both LUAD and LUSC.

In conclusion, our results demonstrate a unifying oncogenic role for ΔNp63 in non-small cell lung cancer consisting of the modulation of the enhancer landscape in lung cancer stem cells. This modulation ultimately leads to the expression of key drivers of LUSC and LUAD, whose inhibition may represent a novel therapeutic approach to treat these highly deadly tumour types.

## Methods

**Cell lines and culture conditions**. The $Kras^{G12D/+}$ expressing human lung cancer cell lines (H520, H2170, H358, H1944 and H2009) were cultured in RPMI medium supplemented with 1% L-glutamine, 1% penicillin/streptomycin, and 10% foetal bovine serum. 293 T cells were cultured in DMEM medium supplemented with 1% L-glutamine, 1% penicillin/streptomycin, and 10% foetal bovine serum. All the cell lines were authenticated by STR profiling by the MD Anderson Cell Line Authentication Service.

**Mouse models**. $ΔNp63^{fl/fl}$ mice[12], $Rosa^{M/M}$ mice[30] and $Kras^{LSL-G12D/+}$ mice[22] were in a C57BL/6 background and were crossed to obtain the following three mouse strains: (i) $ΔNp63^{fl/fl};Rosa^{M/M}$, (ii) $Kras^{G12D/+};Rosa^{M/M}$, (iii) $ΔNp63^{fl/fl};Kras^{G12D/+};Rosa^{M/M}$. Mice were aged until 6–8 weeks old before being utilised in experiments. Both male and female mice were used at equal ratios. Littermates were randomly assigned to experimental groups. Mice were housed pathogen-free and ventilated cages and allowed free access to irradiated food and autoclaved water ad libitum in a 12 h light/dark cycle, with room temperature at $21 \pm 2\,°C$ and humidity between 45 and 65%. All studies were performed in accordance with established protocols approved by the Institutional Animal Care and Use Committee of MD Anderson Cancer Center and H. Lee Moffitt Cancer Center.

**Intratracheal infection of mice**. Six to eight weeks old mice were anaesthetised with a ketamine/xylazine solution (ketamine 100 mg/kg IP, xylazine 10 mg/kg IP), intubated using a fibre optic illuminated catheter[29], and given $2.5 \times 10^7$ PFU viral dosage suspended in 100 μl of minimal essential media (MEM). CMV-cre and CMV-empty adenovirus was obtained from the Viral Core at the Baylor College of Medicine. At the indicated timepoints, lungs and tracheas were isolated from euthanized mice and preserved in 10% formaldehyde for 24 h, followed by preservation in 70% ethanol for 24 h. The tissues were then processed by the MD Anderson Research Histology Core Lab.

**Immunofluorescence and immunohistochemistry**. IF and IHC was performed overnight keeping the slides in humified chambers[11] and incubating them with the following primary antibodies: ΔNp63 (ab172731, Abcam, 1:500), GFP (ab13970, Abcam, 1:200), Ki67 (ab15580, Abcam, 1:1,000), cytokeratin 5 (ab53121, Abcam, 1:250), acetylated tubulin (T7451, Sigma-Aldrich, 1:200), mucin 5ac (MA1-21907, Thermo Fisher Scientific, 1:100), cleaved caspase 3 (9661, Cell Signaling, 1:200), CC10 (sc-25555, Santa Cruz, 1:200), SPC (sc-7705, Santa Cruz, 1:100) and NGFR (ab8874, Abcam, 1:100).

**Microscopy and image processing**. Bright-field and immunofluorescent images were taken on an Olympus IX83 microscope. Immunohistochemical and immunofluorescence images were quantified using ImageJ cell counting software.

**Scoring of tracheal sections**. Three H&E sections per mouse were analysed using the Olympus Cellsense software. Tracheal epithelial height was measured from the basement membrane to the apex of the membrane. Epithelial separation (ES) was defined as the length of epithelium where the basement membrane was separated.

**Lung tumour grading**. H&E sections of mouse lungs were examined, and lung lesions were graded[22] into atypical adenomatous hyperplasia (AAH), grade 1, and grade 2+. For each mouse, three H&E sections of five lung lobes taken 100 μm apart were analysed.

**Polidocanol administration**. Fifty microlitres of 2% polidocanol/PBS was administered to the mouse trachea by a catheter inserted into the trachea[34].

**Mouse tumour xenografts**. About $5 \times 10^6$ human lung cancer cells were mixed in 1:1 ratio with growth factor reduced matrigel (Corning) in a total volume of 200 μl to inject per flank in 6–8 weeks old male athymic $nu/nu$ mice. Tumours were measured weekly with callipers until the final volume reached around 2 cm$^3$. Mice were euthanized and the tumours were excised, measured, fixed in 10% formaldehyde and submitted for histological processing.

**Soft agar colony assay of tumour cell lines**. Human lung cancer cells were transfected with siRNAs for ΔNp63[16,55] and pCMV6-Myc tagged BCL9L (RC218806, OriGene Technologies). siNon-Targeting (siC001, Sigma) and pCMV6 (PS100001, OriGene Technologies) were used as negative controls, respectively. After 24 h, the cells were trypsinized and resuspended in a top layer of their culture medium with 0.3% agarose (BP160, Fisher) at $2 \times 10^5$ cells per well in triplicate in six-well plates and plated on a bottom layer of culture medium containing 1% agarose. The medium was changed every 2 days. After 3 weeks the colonies were counted with a 20X objective on a Zeiss Observer.Z1 microscope.

**Western blot assay**. Human cancer cells were lysed and western blot was performed[16] using the following primary antibodies were used: ΔNp63, (619002, BioLegend, 1:500), BCL9L (PA5-61946, Thermo Fisher, 1:1000), Myc-tag (2276, Cell Signaling, 1:2000) and Actin (4967 S, Cell Signaling, 1:7500).

**Isolation of mouse basal stem cells**. The tracheas from $ΔNp63^{fl/fl};Rosa^{M/M}$ mice were isolated and digested with papain at 37 °C[3]. Basal cells were then cultured on mitomycin treated J2-3T3 feeders for 7 days in MTEC media with rock inhibitor (ab120129, Abcam).

**In vitro adenoviral infection of mouse lung cells**. Cells were infected at a multiplicity of infection (MOI) of 100. Adenovirus (adeno-empty and adeno-cre) was added to cells in a total volume of 2 ml and incubated at 37 °C for 1 h, agitating every 15 min. Cells were then washed 2x with PBS.

**Proliferation assay of lung basal stem cell**. Basal cells were plated on a bed of mitomycin treated J2-3T3 feeders. 5-ethynyl-2′-deoxyuridine (EdU) was added to the media for 24 h. Cells were fixed with paraformaldehyde and developed using the Click-it EdU Alexa Fluor 488 imaging kit (Invitrogen). Data were analysed on the Nexcelom Celigo.

**Sphere formation assay of basal stem cells**. Basal cells in single-cell suspension were seeded into 100 μl of 1:1 growth factor reduced matrigel:MTEC medium in a 96-well plate. About 100 μl of MTEC plus medium was seeded on top of the matrigel. The medium was changed every 2 days. Cells were grown for 1 week. Matrigel was dissolved with cell recovery solution (Corning) and the plate was placed at 4 °C for 2 h to allow the spheres to sink to the bottom of the plate. The plate was then scanned and analysed on the Nexcelom Celigo.

**Differentiation assay of basal stem cells**. Basal cells were seeded into 1:1 GFR matrigel: MTEC plus medium into a 24 well plate insert. MTEC plus medium in the well was changed every 2 days and rock inhibitor was omitted after day 4 to allow for differentiation. After 20 days of growth, the medium was removed and replaced with 4% paraformaldehyde for 24 h of fixation. The fixed matrigel sample was removed and embedded in Histogel, followed by histological processing.

**Apoptosis assay of basal stem cells**. Basal cells were plated on top of mitomycin treated J2-3T3 feeders. Using the Annexin V/Dead Cell Apoptosis Kit (Thermo Fisher), cells were incubated with reagent and subsequently fixed. Analysis was conducted using the Nexcelom Celigo for imaging.

**Isolation of distal lung stem cells**. BASCs and AT2 cells were isolated from $ΔNp63^{fl/fl};Rosa^{M/M}$ mice[6,56]. After digestion with collagenase and dispase, cells were stained with fluorescent antibodies: CD31-APC (551262, BD), CD45-APC (559864, BD), EpCAM-PE-Cy7 (118216, BioLegend), SCA-1-APC-Cy7 (560654, BD). Cells were sorted in a MoFlo Astrios cell sorter. AT2 cells are contained in the CD31$^{neg}$, CD45$^{neg}$, EpCAM$^{pos}$ and Sca-1$^{neg}$ population, and BASCs in the CD31$^{neg}$, CD45$^{neg}$, EpCAM$^{pos}$ and Sca-1$^{pos}$ population[57].

**Colony formation assay of distal lung stem cells**. BASC and AT2 cells were plated in a gelatin-coated 96-well plate mixed with primary mouse lung endothelial

cells in a ratio of 1000 stem cells to $2 \times 10^6$ endothelial cells per plate. The medium was changed every 2 days. The plate was analysed on the Nexcelom Celigo after 1 week of culture[6] using the colony counting algorithm.

**RNA isolation**. RNA was extracted from a minimum of $1 \times 10^6$ primary cell culture or cancer cell line culture using trizol extraction and the Pure Link RNA mini kit (Ambion).

**RNA sequencing**. Approximately 5 μg of polyA+RNA was used to construct RNA-Seq libraries using the standard Illumina protocol. Mouse and human mRNA sequencing yielded 20–40 million read pairs for each sample. The mouse mRNA-Seq reads were mapped using TopHat v2.0.12 onto the mouse genome mm10 (GRCm38) assembly. Then, mapped reads were assembled using cufflinks v2.2.1 to calculate the fragments per kilobase of transcript per million mapped reads (FPKM). A combined profile of all samples was computed. Principal component analysis was executed using the implementation within the R statistical analysis system. Hierarchical clustering of samples was executed by first computing the symmetrical sample distance matrix using the Pearson correlation between mRNA profiles as a metric, supervised sample analysis was performed using the $t$-test statistics, and heatmaps were generated using the heatmap.2 package in R. Enriched pathways were determined using the hypergeometric distribution, with significance achieved for FDR-adjusted $P < 0.05$, and are reported in the Supplementary dataset 1.

**Low cell number ChIP-sequencing**. $ΔNp63^{fl/fl}$ and $ΔNp63^{Δ/Δ}$ cells were pelleted and snap-frozen using liquid nitrogen. Low cell number ChIP-seq for H3K27ac (39193, Active Motif), RNA polymerase 2 (39097, Active Motif), and ΔNp63 (sc-8609, Santa Cruz) was performed by Active Motif. Sequencing was analysed with MACS[58] to call peaks with $q < 0.05$ and mapped using bowtie2 against the mouse genome build UCSC mm10. ChIP-Seq signal maps were generated using BEDTools[59] and visualised using the Integrative Genome Viewer (IGV)[60]. Enhancer candidates were determined using HOMER[61]. Enriched motifs at peaks were determined using HOMER[61]. ChIP-Seq signals at top 2000 enhancer candidates plotted as heatmaps using deepTools[62] scaled for enhancer length and summaries over all enhancer candidates were computed as average values over all enhancer candidates and plotted. Overlap with known enhancers was determined using BEDTools[59] against the Fantom 5 enhancer catalogue[41] and the Mouse ENCODE lung enhancers[42], and it is reported in Supplementary dataset 3. The enhancer-associated genes are listed in Supplementary datasets 2 and 4.

For the validation of the selected genes, cellular proteins were cross-linked to DNA using 1% formaldehyde and chromatin was isolated and sonicated in ChIP buffer (20 mM Tris-HCl pH 7.5, 100 mM NaCl, 1 mM EDTA, 0.5% NP40, 0.5% sodium deoxycholate, 0.1% SDS)[63]. Each ChIP was performed in triplicate using 2 μg of either H3K27ac (39193, Active Motif), RNA polymerase 2 (39097, Active Motif) or ΔNp63 (sc-8609, Santa Cruz). As negative controls for the immunoprecipitation, IgG purified from mouse serum (sc-2025, Santa Cruz) and rabbit serum (sc-2027, Santa Cruz) were used. The presence of H3K27ac, RNA polymerase 2 and ΔNp63 was analysed by qRT-PCR with the primers listed in Supplementary dataset 7.

**Quantitative real time PCR**. Total RNA was prepared using TRIzol reagent (Invitrogen). For gene expression analysis, complementary DNA was synthesised from 5 μg of total RNA using the SuperScript II First-Strand Synthesis Kit (Invitrogen) according to the manufacturer's protocol followed by qRT-PCR using Bio-Rad SYBR green master mix. The qRT-PCR data were analysed using the ΔΔCT analysis. The utilised primers are listed in Supplementary dataset 8.

**dCas9 assay**. Experiments were performed in 293 T cells as previously reported[53]. Briefly, four sgRNA per genomic site were designed using the Broad sgRNA design tool and cloned into the pLX-sgRNA vector. Plasmids for dCas9, dCas9-p300 and dCas9-ΔNp63 were utilised in a 3:1 ratio to sgRNA vectors. All plasmids were transfected using Lipofectamine 2000. The cells were incubated for 48 h before collection of the RNA. The utilised primers are listed in Supplementary dataset 9.

**The Cancer Genome Atlas (TCGA) lung adenocarcinoma (LUAD) and lung squamous cell carcinoma (LUSC) datasets**. Clinical, mutation data and normalised RNA-seq expression data (RSEM—Batch normalised from Illumina HiSeq_RNASeqV2) of TCGA LUAD and LUSC patients were obtained from the cBioPortal (https://www.cbioportal.org) pan-cancer cohort. Out of 566 LUAD patients in the dataset, 510 patients have RNA-seq data where 501 patients had complete overall survival information and 508 samples had AJCC pathologic tumour stage information. Similarly, a total of 487 LUSC patients were found in the clinical data, with 484 of those were available in the RNA-seq expression data, and 478 patients with complete overall survival information availability were used for overall survival analysis.

**Moffitt LUAD and LUSC datasets**. For the Moffitt LUAD cohort, gene expression profiles of 442 patients were downloaded from GEO (GSE72094). This dataset was

profiled by the Rosetta/Merck Human RSTA Custom Affymetrix 2.0 microarray and the data were normalised by IRON as previously described[64]. About 398 patients have the survival information and were used for the survival analysis. For the Moffitt LUSC cohort, 116 patients were profiled by RNA-seq and 108 had clinical outcome data as previously described[65].

**Overall survival analysis**. Median gene expression of BCL9L was used as the cut-point to classify high and low expressed groups. Kaplan–Meier overall survival analysis was performed between the high and low gene expression group using 'KapalanMeierFitter' function in 'lifelines' Python package (version 0.25.4). *P* values were obtained using log rank test through 'multivariate_logrank_test' function in the same package and *P* < 0.05 was considered statistically significant in this analysis.

**Statistical analyses and reproducibility**. All the experiments are representative of at least three independent replicates. Data collection was performed with Microsoft Excel and data analysis was performed using GraphPad Prism. Gels, blots and images are representative of three independent experiments giving similar results.

**Reporting summary**. Further information on research design is available in the Nature Research Reporting Summary linked to this article.

## Data availability

Publicly available data used in this paper were obtained from UCSC (http://genome.ucsc.edu/cgibin/hgGateway) and The Cancer Genome Atlas (TCGA) lung adenocarcinoma dataset (http://www.cancer.gov/about-nci/organization/ccg/research/structural-genomics/tcga). The RNA-seq data and the ChIP-seq data were deposited to NCBI Gene Expression Omnibus (GEO) repository (series GSE131671). All the imaging data supporting the findings of this study are available from the corresponding authors upon reasonable request. The source data for all the other results are provided as Source Data file with this paper. Source data are provided with this paper.

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

## Acknowledgements
We thank C. Kim and M. Paschini for generously sharing protocols and cell lines for AT2 and BASC isolation and culture. We thank B. Gomperts for sharing polidocanol protocols. We thank C. Liu and J. Kurie for intratracheal instillation training and P. Raulji for assistance with the maintenance of mouse colonies. This work was supported by R35CA197452 to ERF, R01 CA207098, and R01 CA207109 to MGL, and CPRIT RP140271 to ERF and MGL. ERF is a National Cancer Institute Outstanding Investi-gator, Moffitt Distinguished Scholar, and Scholar of the Leukaemia and Lymphoma Society, the Rita Allen Foundation, and the V Foundation for Cancer Research. MN is a Scholar of the Cancer Prevention Research Institute of Texas-Translational Research in Multidisciplinary Programme. SJW was supported by T32 in Molecular Genetics of Cancer (T32 CA009299), the Schissler Foundation for Translational Studies of Common Human Diseases, and the Andrew Sowell-Wade Huggins Scholarship in Cancer Research. KR and CC have been supported in part by NIH P30 shared resource grant CA125123, CPRIT RP200504, NIEHS P30 Center grant ES030285, and NIEHS P42 ES027725. KL was supported by T32 in Integrated Programme in Cancer and Data Science (T32 CA233399).

## Author contributions
M.N., S.J.W. and E.R.F. conceived the study, designed experiments and analysed data. M.N., S.J.W., B.L.G., R.C. and S.D. performed experiments. C.C., H.A.A., K.L., K.R. and A.C.T. conducted bioinformatics analyses. M.N., S.J.W., M.G.L. and E.R.F. wrote the paper. All authors discussed the paper and commented on the manuscript.

## Competing interests
The authors declare no competing interests.

## Additional information

**Peer review information** *Nature Communications* thanks De-chen Lin, Giovanni Blandino and the other anonymous reviewer(s) for their contribution to the peer review this work. Peer reviewer reports are available.

