## [Peer Review File · Nature Communications]

Reviewers' Comments:

Reviewer #1:

Remarks to the Author:

In this manuscript, the authors tested the function of $\Delta Np63$ by knocking it out in a Kras-driven GEMM LUAD model as well as a xenograft model of LUSC. They found decreased tumor development and growth in both models. They went on and investigated $\Delta Np63$ function in normal lung basal cells and AT2 cells, which are proposed cell-of-origins for LUSC and LUAD, respectively. They also studied the epigenetic changes after deleting of $\Delta Np63$ in these two cell types. Overall, the study was systematic and revealed new insights into the role of $\Delta Np63$ in both lung cancer types as well as the cell-of-origin. However, careful review found a number of weaknesses and major concerns from experimental design, data interpretation, as well as scientific rigor, as outlined below:

1) No validation was provided for the knockout of $\Delta Np63$ in the Kras LUAD mouse model. Which cell type does $\Delta Np63$ get deleted? I was assuming $\Delta Np63$ was knocked out across many types of lung cells? If that was the case, then the authors should at least discuss non-autonomous mechanisms which might play a role here. In other words, $\Delta Np63$ in other cell type might affect the initiation of adenoma and LUAD from AT2 cells.

Relatedly, the authors took a GEMM to study $\Delta Np63$ for LUAD, but only investigated the LUSC subtype using a xenograft model, which is inadequate and hard to compare. It would add more enthusiasm and value to this work if a similar GEMM is utilized for LUSC.

More detailed analyses of the Kras model is need, since this Kras model gives rise to both adenoma and LUAD. For example, did $\Delta Np63$ deletion affect the survival of these mice? did $\Delta Np63$ deletion regulate the differentiation status of the carcinoma cells?

It is known that $\Delta Np63$ is expressed at low levels of LUAD tumors. Thus, for the human LUAD cell line experiments (Fig.1i), western blot validation is required.

Again, in the $\Delta Np63\Delta/\Delta; Rosa\Delta/\Delta$ mouse model (Fig.2), genetic validation of the $\Delta Np63$ knockout efficiency is required.

One major issue with the epigenomic data is in Fig.7A, while the authors identified over 10,000 super-enhancer genes in basal cells, in AT cells they only found 102. Both data are very unusual. There are perhaps over hundreds of cell lines and/or primary cells which have been profiled for super-enhancers annotation, and the number of super-enhancers ranges 500-1500 per cell type. This brings questions to the data quality of these Chip-Seq. How many peaks were identified? at what statistical significance? Another puzzling point is, the number of super-enhancer genes in Fig.7A and Fig. 6B is inconsistent.

The authors described that they performed low-cell number ChIP-seq analysis using an antibody directed against $\Delta Np63$. However, no $\Delta Np63$ Chip-seq peaks were shown. The representative $\Delta Np63$ peaks need to be provided to assess the chip-seq quality.

In the pathway analysis (Fig.5B), instead of raw p value, multi-test correction is required.

Fig.5E is not an ideal presentation of the comparison between control and $\Delta Np63$ knockout. There are a couple ways to show this. For example, they can just show the differential peak signals of these super-enhancers between control and $\Delta Np63$ knockout, rank ordered by peak intensity. Moreover, is X axis showing only super-enhancers? Since the most common way to define super-enhancer is to use the inflection point, it seems that it is more intuitive to show all enhancers (including both typical- and super-enhancers). And why are both groups disappearing at 1000? It seems to be an artificial hard cut-off? Similarly, Fig.6B also got cut out at 2000. But I don't think this is common done for super-enhancer annotation.

Another important question is, what is the rationale to restrict the analysis on super-enhancer only? $\Delta Np63$ is known to regulate both typical- and super-enhancers. The reason to exclude the

analysis of typical enhancers need to be supported by data.

There is very trivial difference of the signal intensity between control and Δ Np63 in Fig.6D and E. The change in Fig.5H is also moderate. Do these changes meet both the statistical test (e.g., certain q value?) as well as certain fold change?

Also, I do not understand the motif bar in Fig.5G. Δ Np63 motif is just around 20-30 bp, and why the motif bar is as long as 5kb?

The authors focused strongly on KRT5 and BCL9 genes, but both of them ranked quite low in the super-enhancer list. I understand that they were discovered initially from Fig.7A, but again, the result of Fig.7A needs to be re-visited. Moreover, as mentioned above, the IGV tracks do not show evident change of peak intensity of BCL9 in either cell type.

Reviewer #2:

Remarks to the Author:

In the present manuscript the authors find that Δ Np63 plays a pivotal role in the maintenance of stemness in the normal lung and in driving an oncogenic transcriptional program. This takes place through the formation of specific chromatin structures (super-enhancers) involving specific basal cell identity-genes. The authors find that this mechanism is maintained both in mouse KRAS-driven LUAD as well as in lung AT2 and basal cells. These are progenitor cells with stem-like properties, necessary for normal lung regeneration and thought to be cells-of-origin of LUSC and LUAD, respectively. The authors also translate their findings in human datasets of LUAD and LUSC, and in human cell lines, finding KRT5, ETV5 and BCL9L genes as important stemness genes regulated by Δ Np63. In particular, in vitro they find that BCL9L and KRT5 are Δ Np63 targets critical for the maintenance of LUSC, while BCL9L and ETV5 are Δ Np63 targets critical for the maintenance of LUAD. The authors further demonstrate that BCL9L depletion both in LUAD and in LUSC cell lines reduces tumorigenesis in vivo in mice xenografts. Finally, the authors find a prognostic role for BCL9L expression both in LUAD and LUSC human casuistries. The work is rather novel, the experiments are well performed and the related results are of high quality.

The listed below specific comments need to be addressed for further evaluation in Nature Communications:

1. In figure 1A and 1B, the authors need to indicate how many mice were used for the analysis of tumor lesions. Same comment for Figure 1G, H and in general for all the figures in which the number of samples is not indicated.
2. In figure 1I, authors only describe colony assay for H1944 and H2009. They should mention also H358.
3. For a better evaluation of colony assay experiment in the different cell lines (Fig. 1I, J, K), would it be possible to indicate the percentage of colonies respect to relative control instead of absolute number of colonies?
4. In describing Figure 3, the authors mention Ac-tubulin and Mucin5 as markers of ciliated and goblet cells, respectively, and they also mention other differentiation markers. However, in the figure 3 panels there are no other differentiation markers analyzed. Authors need to solve this issue.
5. Page 10, lines 197-200 are not clear. What does it mean that the proliferation of the tracheal epithelium at 1 month after Δ Np63 depletion is not due to compensatory de-differentiation of ciliated and goblet cells? From the immunofluorescence experiment shown in fig. 3B-M, after epithelial damage there is loss of differentiated cells in Δ Np63 depleted epithelium. However, the sentence "the proliferation of Δ Np63 Δ/Δ ;Rosa Δ/Δ tracheal epithelium at the 1-month time point post adenoviral cre injection (see Fig. 2j) is primarily due to basal cell proliferation and not a compensatory dedifferentiation of the ciliated or goblet cells." is not clear.
6. For Figure 4A-D, authors should provide some representative images, even in Supplementary material. In the legend relative to figure 4E, authors should explain how do they mark the tracheosphere (EdU)? Same comment for figure 4T: what the red signal stands for?
7. In figure 5G and 5H, it seems unrealistic that a Δ Np63 motif would be 1-7kb long. Maybe the scale of the picture is wrong? Moreover, a different acetylation pattern between control cells and

Δ Np63 cells seems striking only for KRT5 and BCL9L genes in tracheal cells (Fig. 5G, H), while in distal lung cells it is difficult to appreciate a difference in the acetylation patterns of ETV5 and BCL9L genes with or without Δ Np63. This needs to be mentioned and experimentally clarified.

8. The authors need to analyze the presence of a positive correlation between Δ Np63 and BCL9L in LUAD and LUSC cohorts.

9. Rescue experiments in vitro showing that BCL9L overexpression in Δ Np63 cells may rescue oncogenic properties (stemness, anchorage-independent growth etc) are required.

Reviewer #3:

Remarks to the Author:

Summary

In this manuscript, Napoli et al. dissect the role of DNp63 in NSCLC, including adenocarcinoma (LUAD) and squamous cell carcinoma (LUSC). The conditional allele that enables specific ablation of DNp63 in vivo and in vitro is a major strength of the study. Overall, the data associated with LUSC and its cell(s) of origin is convincing and either confirms or extends previous observations about the role of DNp63 in squamous cell carcinoma arising in different tissues. In particular, the analysis of DNp63 in tracheal basal cells is quite intriguing. However, the data on LUAD do not fully support the authors' conclusions and need to be expanded to adequately delineate the role of DNp63 in this disease.

Major points

1. The data in Figure 1A clearly support a role for DNp63 at some stage of Kras-driven lung tumorigenesis. However, there are concerns with other panels in this figure as outlined below.

2. Figure 1C. The IHC for DNp63 shown does not support the statement "Tumors from KrasG12D/+ mice stained positive for Δ Np63". The signal to noise is quite low in this image and it is difficult to believe that all tumor cells are positive. If a subset of cells are positive, it is hard to distinguish them. Additional characterization and precise description of DNp63 expression in control tumors is needed. Options could include: using a different antibody for IHC (such as the monoclonal antibody BC28, which has been used in multiple publications on human and mouse FFPE tissue); including positive control normal cells (such as basal cells) from the same slide to illustrate relative staining levels; and evaluation of recent single cell analyses of K and KP tumors (DOI: 10.1016/j.ccell.2020.06.012) to quantitate expression of DNp63 in those datasets. Since most human LUAD are thought to be DNp63-negative (with rare exceptions), characterization of DNp63 expression in this model needs to be very clear, particularly if the conclusion is that DNp63 expression in the mouse model of LUAD differs substantially from human LUAD.

3. Figures 1I/J-M, V: siRNA and shRNA experiments are difficult to interpret in the absence of information (immunoblotting) on baseline expression of DNp63 and the reduction in protein levels. (Specific nuclear positivity is difficult to identify in the IHC in 1P/T.) The use of a single shRNA against DNp63 also makes the data less robust. At a minimum, it would be nice to know if this shRNA has no effect in cell lines lacking DNp63.

4. Figures 2U-B' do not seem to be described in the manuscript.

5. Figure 4. Loss of DNp63 expression after Ad-Cre is not documented anywhere in the figure. This is particularly important for AT2/BASC studies, as these cells are not anticipated to have high levels of DNp63 at baseline. Relative levels of DNp63 in the different cell types are important to document.

6. "BCL9L is critical for maintenance of LUSC and LUAD" is a very broad claim to make when a single cell line of each tumor type has been evaluated. More cell lines and/or mining of public data (like DepMap) are needed. Were these experiments performed in the other LUAD cell lines from figure 1, all of which appear sensitive to DNp63 knockdown? Description of cell line genetics is warranted (how does driver mutation status compare with the GEMM?). As in figure 1, the use of a single shRNA and the absence of immunoblotting for target proteins is an issue. There is also no

data directly demonstrating that BCL9L mediates the effects of DNp63 (rescue of shDNp63 by exogenous BCL9L, for example).

Minor points

1. "we assessed a greater than 90% recombination in the trachea and distal lung assessed by GFP expression from the ROSA allele". This is not a very precise description...90% of all cells? Or of all epithelial cells? It should be described in more detail and documented if essential to the manuscript. Or it could potentially be removed.
2. Figure 1C. I have no doubt that there were tumors in the KrasG12D model, but the H&E image here contains a cluster of lymphocytes. A different image of a tumor should be taken.
3. Figure 1D. The cells here appear to mainly include a junction between an airway and alveoli. Ideally, Figures 1C-F would compare tumors of the same grade (grade 1?) from each genotype so that there is an apples to apples comparison. Depicting AAH from each genotype would also be helpful in understanding which stage of tumorigenesis DNp63 expression is most readily detectable.
4. "...suggesting that Δ Np63 serves to maintain the proliferation of distal lung stem cell populations (Fig. 1g,h)." This conclusion is not really supported by the observation.
5. Figure 5G/H – these seem to depict H3K27ac signal mainly at promoters – are there known enhancers for these genes, either in the depicted region or elsewhere.
6. Although it may be beyond the scope of a revision, if the authors have done any experiments with Cre driven by cell type-specific promoters (SPC-Cre, CCSP-Cre), this would be informative.

Response to Reviewers' Comments:

We would like to thank the reviewers for their constructive comments and suggestions. We have addressed all the points raised by the reviewers and believe that the manuscript is of high significance to the cancer stem cell and super-enhancer fields. We now provide additional evidence of the oncogenic activities of *BCL9L* as a $\Delta Np63$ regulated super-enhancer associated gene. In this resubmission, we include data demonstrating the regulation of *BCL9L* expression by $\Delta Np63$ in both human lung adenocarcinoma and lung squamous cell carcinoma cell lines. More importantly, we also unveiled the crucial role of *BCL9L* as a mediator of the pro-tumour functions of $\Delta Np63$ in non-small cell lung cancer. Indeed, overexpression of *BCL9L* is sufficient to rescue the colony formation caused by the downregulation of $\Delta Np63$ in a panel of human lung cancer cells. Our specific point-by-point response to each comment is below in boldface type.

Reviewers' comments:

Reviewer #1 (Remarks to the Author):

In this manuscript, the authors tested the function of $\Delta Np63$ by knocking it out in a Kras-driven GEMM LUAD model as well as a xenograft model of LUSC. They found decreased tumor development and growth in both models. They went on and investigated $\Delta Np63$ function in normal lung basal cells and AT2 cells, which are proposed cell-of-origins for LUSC and LUAD, respectively. They also studied the epigenetic changes after deleting of $\Delta Np63$ in these two cell types. Overall, the study was systematic and revealed new insights into the role of $\Delta Np63$ in both lung cancer types as well as the cell-of-origin.

We would like to thank the reviewer for deeming our study systematic and appreciating the new roles for $\Delta Np63$ in lung cancers and their cells of origin that we unveiled.

However, careful review found a number of weaknesses and major concerns from experimental design, data interpretation, as well as scientific rigor, as outlined below:

1) No validation was provided for the knockout of $\Delta Np63$ in the *Kras* LUAD mouse model. Which cell type does $\Delta Np63$ get deleted? I was assuming $\Delta Np63$ was knocked out across many types of lung cells? If that was the case, then the authors should at least discuss non-autonomous mechanisms which might play a role here. In other words, $\Delta Np63$ in other cell type might affect the initiation of adenoma and LUAD from AT2 cells.

We intercrossed our mice with a *Rosa* reporter (*Rosa^{M/M}*), which allows for assessment of Cre-recombination via GFP expression. We have included a representative image of the staining for GFP and tdTomato in the tracheal epithelium and distal lungs isolated from $\Delta Np63^{fl/fl};Kras^{LSL-G12D/+};Rosa^{M/M}$ and $\Delta Np63^{\Delta/\Delta};Kras^{G12D/+};Rosa^{\Delta/\Delta}$ mice collected at 20 weeks post-infection with adenoviral cre (Figure 1 and also included as new Supplementary Fig. 1a). As shown in the figure, we observed a greater than 90% recombination in both the basal cells of the trachea and the distal lungs of $\Delta Np63^{\Delta/\Delta};Kras^{G12D/+};Rosa^{\Delta/\Delta}$ mice as assessed by GFP expression from the *Rosa* allele.

Figure 1. Representative image of the staining for GFP and tdTomato in tracheal epithelium and distal lungs isolated of the indicated genotypes. DAPI was used as a counterstaining. Scale bars equal 100 μ m.

As pointed out by the reviewer, we stated in the methods that CMV-cre adenovirus was administered intratracheally. Given the nature of the promoter used, multiple cell types may undergo recombination beyond the epithelial cells, including stromal and immune system cells. It is important to note that our *in vitro* data using isolated basal cells, AT2, and BASC cells clearly demonstrate that $\Delta Np63$ is required for self-renewal and terminal differentiation of these lung cell populations in a cell autonomous manner (see Figure 4); we cannot completely exclude that any additional non-autonomous effects may take place *in vivo*, which will be the focus of future studies.

2) Relatedly, the authors took a GEMM to study $\Delta Np63$ for LUAD, but only investigated the LUSC subtype using a xenograft model, which is inadequate and hard to compare. It would add more enthusiasm and value to this work if a similar GEMM is utilized for LUSC.

In contrast to lung adenocarcinoma and its well-established Kras-driven GEMMs, lung squamous cell carcinomas lack GEMMs that can faithfully recapitulate tumour type (as recently reviewed by Hynds R.E. *et al.*, Open Biol. 2021 Jan;11(1):200247. doi: 10.1098/rsob.200247. Epub 2021 Jan 13. PMID: 33435818). The lack of these GEMM models formed the basis of our decision to analyse the contribution of $\Delta Np63$ in lung squamous cell carcinomas by utilizing xenograft models instead. In addition, these results in xenograft models further demonstrate the cell autonomous effects of $\Delta Np63$.

3) More detailed analyses of the Kras model is need, since this Kras model gives rise to both adenoma and LUAD. For example, did $\Delta Np63$ deletion affect the survival of these mice? did $\Delta Np63$ deletion regulate the differentiation status of the carcinoma cells?

The $Kras^{G12D/+}$ mice develop both lung adenomas and lung adenocarcinomas (LUAD) up to grade 4 if the tumours are analysed at 30 weeks post-infection with adenoviral cre (Jackson E.L. *et al.*, Genes Dev. 2001 Dec 15;15(24):3243-8. doi: 10.1101/gad.943001. PMID: 11751630). Since we wanted to characterize the role of $\Delta Np63$ in LUAD initiation, we decided to euthanize both $Kras^{G12D/+};Rosa^{A/A}$ and $\Delta Np63^{A/A};Kras^{G12D/+};Rosa^{A/A}$ mice at an earlier timepoint (i.e. 20 weeks post-infection with adenoviral cre). Even though we did not

let the mice reach their natural endpoint (i.e. overall survival was not measured), our analysis of the tumour grade clearly shows that almost none of the $\Delta Np63^{A/A};Kras^{G12D/+};Rosa^{A/A}$ tumours was of a grade higher than 1 (see Fig. 1b).

4) It is known that $\Delta Np63$ is expressed at low levels of LUAD tumors. Thus, for the human LUAD cell line experiments (Fig.1i), western blot validation is required.

As requested by the reviewer, we have now included a western blot validation of the $\Delta Np63$ downregulation in the human lung cancer cell experiments. As also recommended by reviewer #3 (see point #3), we have now included the data showing the downregulation of $\Delta Np63$ by 2 independent sequences previously used by our laboratory (Napoli M. *et al.*, Cancer Cell. 2016 Jun 13;29(6):874-888. doi: 10.1016/j.ccell.2016.04.016. PMID: 27300436; and Bui N.H.B. *et al.*, Cancer Res. 2020 Jul 1;80(13):2833-2847. doi: 10.1158/0008-5472.CAN-19-2733. Epub 2020 Apr 20. PMID: 32312834). The most efficient of the 2 sequences (indicated in the manuscript as si $\Delta Np63$ #1 and sh $\Delta Np63$ #1) is the one that was utilized for the experiments with LUAD cells (see Fig. 1e) and LUSC cells (see Fig. 1f-r). The western blot analysis for $\Delta Np63$ (now included in the manuscript as new Supplementary Fig. 1b,c) is also shown below as Figure 2.

Figure 2. Representative western blot analysis for $\Delta Np63$ in the indicated siRNA transfected LUAD cell lines (a) and shRNA infected LUSC cell lines (b). Actin was used as a loading control.

5) Again, in the $\Delta Np63^{\Delta/\Delta}; Rosa^{\Delta/\Delta}$ mouse model (Fig.2), genetic validation of the $\Delta Np63$ knockout efficiency is required.

The *Rosa* allele present in the $\Delta Np63^{fl/fl}; Rosa^{fl/fl}$ mice provided us with the great advantage of being able to easily monitor the recombination occurring after the administration of adenoviral cre. Indeed, after the recombination, the *Rosa* allele expresses GFP that otherwise is silenced (Muzumdar M.D. *et al.*, *Genesis*. 2007 Sep;45(9):593-605. doi: 10.1002/dvg.20335. PMID: 17868096). In Figure 2 – but the same also applies to the data shown in both Fig. 3 and 4 in the manuscript – we performed a staining for GFP to show that the recombination is occurring and thus $\Delta Np63$ is deleted in the $\Delta Np63^{\Delta/\Delta}; Rosa^{\Delta/\Delta}$ cells and tissues, while no GFP is detected and thus $\Delta Np63$ is still expressed in the $\Delta Np63^{fl/fl}; Rosa^{fl/fl}$ counterparts.

6) One major issue with the epigenomic data is in Fig.7A, while the authors identified over 10,000 super-enhancer genes in basal cells, in AT cells they only found 102. Both data are very unusual. There are perhaps over hundreds of cell lines and/or primary cells which have been profiled for super-enhancers annotation, and the number of super-enhancers ranges 500-1500 per cell type. This brings questions to the data quality of these Chip-Seq. How many peaks were identified? at what statistical significance? Another puzzling point is, the number of super-enhancer genes in Fig.7A and Fig. 6B is inconsistent.

Based on the reviewer's comments on Fig. 7a and on the difference between the rankings of the enhancers shown in Fig. 5e and 6b (see point #9 below), we repeated our analysis by ranking the top 2000 enhancers identified in the basal cells similarly to what we had previously done with the data obtained in AT2 cells (see Figure 3 below, also included in the manuscript as new Fig. 5e and the previously present Fig. 6b).

Figure 3. Ranking of top 2000 enhancers based on H3K27ac ChIP-seq in $\Delta Np63^{fl/fl}; Rosa^{M/M}$ (blue) and $\Delta Np63^{Δ/Δ}; Rosa^{Δ/Δ}$ (red) basal cells (a) and AT2 cells (b).

These lists of the top 2000 enhancers and their respective scores are now included as new Supplementary Tables 2 and 4, while the total number of peaks identified in AT2 and basal cells is now included in the new Supplementary Table 3. The cut-off of top 2000 enhancers allowed us to retain *Bcl9l* – a gene that we now proved to be essential to mediate the oncogenic activities of $\Delta Np63$ in both LUAD and LUSC – among the genes regulated by $\Delta Np63$ in both basal cells and AT2 cells, which are listed in the new Supplementary Table 5. We therefore updated Fig. 7a by including the genes associated with the top 2000 enhancers regulated by $\Delta Np63$ in these two cell types (see Figure 4 below, also included in the manuscript as new Fig. 7a).

Figure 4. Venn diagram overlaying the genes associated with the top 2000 $\Delta Np63$ -regulated enhancers in AT2 and basal cells.

7) *The authors described that they performed low-cell number ChIP-seq analysis using an antibody directed against Δ Np63. However, no Δ Np63 Chip-seq peaks were shown. The representative Δ Np63 peaks need to be provided to assess the chip-seq quality.*

Indeed, low-cell number ChIP-seq was performed due to the technical challenge of obtaining primary cells from mice, especially AT2 cells, that express Δ Np63 at low levels. Therefore, the Δ Np63 regulated regions were identified by searching for the Δ Np63 motif in the H3K27ac peaks as shown in Fig. 5g,h and Fig. 6d,e. The regulation of the expression of selected genes by Δ Np63 was validated and confirmed by qRT-PCR in Δ Np63^{fl/fl};Rosa^{fl/fl} and Δ Np63^{Δ/Δ};Rosa^{Δ/Δ} cells (Fig. 5f and 6c). Furthermore, the recruitment of Δ Np63 and its effect in promoting H3K27ac levels to regulate these genes was demonstrated in 2 distinct human cell systems: an exogenous model utilizing dCas9 technology (Fig. 7c-e) and an endogenous model of lung cancer (Fig. 7f-h).

8) *In the pathway analysis (Fig.5B), instead of raw p value, multi-test correction is required.*

We thank the reviewer for raising this point. We have now clarified in the methods that the “enriched pathways were determined using the hypergeometric distribution, with significance achieved for FDR-adjusted $P < 0.05$.” The list of pathways and their respective FDR-adjusted p-values is now included as new Supplementary Table 1.

9) *Fig.5E is not an ideal presentation of the comparison between control and Δ Np63 knockout. There are a couple ways to show this. For example, they can just show the differential peak signals of these super-enhancers between control and Δ Np63 knockout, rank ordered by peak intensity. Moreover, is X axis showing only super-enhancers? Since the most common way to define super-enhancer is to use the inflection point, it seems that it is more intuitive to show all enhancers (including both typical- and super-enhancers). And why are both groups disappearing at 1000? It seems to be an artificial hard cut-off? Similarly, Fig.6B also got cut out at 2000. But I don't think this is common done for super-enhancer annotation.*

As indicated above in our reply to point #6, we have now replaced Fig. 5e with a new version in which the top 2000 Δ Np63-regulated enhancers are depicted similarly to what shown in Fig. 6b (please, see Figure 3 above and the new Fig. 5e included in the manuscript) and explained our rationale for using this cut-off. Additionally, to better show the comparison of the H3K27ac signals in these top 2000 enhancers between the Δ Np63^{fl/fl};Kras^{LSL-G12D/+};Rosa^{M/M} and the Δ Np63^{A/A};Kras^{G12D/+};Rosa^{A/A} basal cells, we have now graphed the plots and the heatmaps of these enhancer regions. As shown in Figure 5, also included as new Supplementary Figure 2a in the manuscript, deletion of Δ Np63 causes an overall reduction in the intensity of the H3K27ac signal in these top 2000 enhancers.

H3K27ac signal at top 2000 enhancers in basal cells

Figure 5. H3K27ac chip-seq signal at the top 2000 enhancers in basal cells of the indicated genotypes. Plots and heatmaps are scaled over the enhancer size and included the 2kb before and after the enhancers.

10) Another important question is, what is the rationale to restrict the analysis on super-enhancer only? Δ Np63 is known to regulate both typical- and super-enhancers. The reason to exclude the analysis of typical enhancers need to be supported by data.

We would like to clarify that we never excluded any typical enhancers from our analyses. To avoid any confusion, we have now revised the manuscript specifying that we focused on the top 2000 Δ Np63-regulated enhancers in basal cells and AT2 cells.

11) There is very trivial difference of the signal intensity between control and Δ Np63 in Fig.6D and E. The change in Fig.5H is also moderate. Do these changes meet both the statistical test (e.g., certain q value?) as well as certain fold change?

The difference in the intensity of all the H3K27ac peaks considered in the manuscript was always significant ($q < 0.05$). This information has now been added to the methods. The scores for the top 2000 Δ Np63-regulated enhancers in basal cells and AT2 cells are listed in the new Supplementary Tables 2 and 4, respectively.

12) Also, I do not understand the motif bar in Fig.5G. Δ Np63 motif is just around 20-30 bp, and why the motif bar is as long as 5kb?

We apologise for the confusion caused by our wording. Yes, a transcription factor motif is only a few base pairs long, and in Fig. 5c we report the Δ Np63 motif that we identified through our ChIP-seq data. In the tracks shown in Fig. 5g,h and Fig. 6d,e we just wanted to highlight which among the H3K27ac peaks also contained this Δ Np63 motif. To clarify that, we have now corrected these figures by labelling these regions as “H3K27ac peaks w/ Δ Np63 motif”.

13) The authors focused strongly on *KRT5* and *BCL9* genes, but both of them ranked quite low in the super-enhancer list. I understand that they were discovered initially from Fig. 7A, but again, the result of Fig. 7A needs to be re-visited. Moreover, as mentioned above, the IGV tracks do not show evident change of peak intensity of *BCL9* in either cell type.

As mentioned in our answer to point 11, the difference in the intensity of these H3K27ac signals had a q value below 0.05 and is statistically significant. More importantly, in our previous version of the manuscript, we showed that Δ Np63 regulates the expression of these genes both in mouse and human cells (see Fig. 5f, Fig. 7d,e) by promoting the H3K27ac levels at the enhancers of these genes (see Fig. 7g,h). Finally, since we identified *BCL9L* as a novel Δ Np63 target gene, we have now provided additional data showing that Δ Np63 regulates the expression of *BCL9L* in a panel of human LUAD and LUSC cell lines (Figure 6, also included in the manuscript as new Supplementary Fig. 3a,e).

Figure 6. qRT-PCR of Δ Np63 and *BCL9L* in the indicated human cancer cell lines transfected with the indicated siRNAs. Data are mean \pm SD, $n = 3$, **** vs. siNT, $P < 0.001$, two-tailed Student's t test.

Reviewer #2 (Remarks to the Author):

In the present manuscript the authors find that $\Delta Np63$ plays a pivotal role in the maintenance of stemness in the normal lung and in driving an oncogenic transcriptional program. This takes place through the formation of specific chromatin structures (super-enhancers) involving specific basal cell identity-genes. The authors find that this mechanism is maintained both in mouse KRAS-driven LUAD as well as in lung AT2 and basal cells. These are progenitor cells with stem-like properties, necessary for normal lung regeneration and thought to be cells-of-origin of LUSC and LUAD, respectively. The authors also translate their findings in human datasets of LUAD and LUSC, and in human cell lines, finding KRT5, ETV5 and BCL9L genes as important stemness genes regulated by $\Delta Np63$. In particular, in vitro they find that BCL9L and KRT5 are $\Delta Np63$ targets critical for the maintenance of LUSC, while BCL9L and ETV5 are $\Delta Np63$ targets critical for the maintenance of LUAD. The authors further demonstrate that BCL9L depletion both in LUAD and in LUSC cell lines reduces tumorigenesis in vivo in mice xenografts. Finally, the authors find a prognostic role for BCL9L expression both in LUAD and LUSC human casuistries.

The work is rather novel, the experiments are well performed and the related results are of high quality.

We would like to thank the reviewer for considering our work novel and for the suggestions provided to improve it.

The listed below specific comments need to be addressed for further evaluation in Nature Communications:

1) In figure 1A and 1B, the authors need to indicate how many mice were used for the analysis of tumor lesions. Same comment for Figure 1G, H and in general for all the figures in which the number of samples is not indicated.

Based on the reviewer's recommendation, we have revisited all the figure legends to include the number of samples utilized.

2) *In figure 1I, authors only describe colony assay for H1944 and H2009. They should mention also H358.*

We thank the reviewer for pointing it out, and we have now corrected the paragraph relative to this figure (now listed in the revised manuscript as Fig. 1e).

3) *For a better evaluation of colony assay experiment in the different cell lines (Fig. 1I, J, K), would it be possible to indicate the percentage of colonies respect to relative control instead of absolute number of colonies?*

We agree with the reviewer that indicating the percentages would allow for an easier comparison across the different cell lines. However, we deem that the current representation with the absolute number of colonies can better highlight the intrinsic oncogenic potential of each cell line.

4) *In describing Figure 3, the authors mention Ac-tubulin and Mucin5 as markers of ciliated and goblet cells, respectively, and they also mention other differentiation markers. However, in the figure 3 panels there are no other differentiation markers analyzed. Authors need to solve this issue.*

We have clarified in the revised manuscript that “the $\Delta Np63^{\Delta/\Delta}; Rosa^{\Delta/\Delta}$ tracheal epithelial cells instead formed a disorganized epithelium of Krt5+ cells and lacked expression of muc5ac and acetylated tubulin”.

5) Page 10, lines 197-200 are not clear. What does it mean that the proliferation of the tracheal epithelium at 1 month after $\Delta Np63$ depletion is not due to compensatory de-differentiation of ciliated and goblet cells? From the immunofluorescence experiment shown in fig. 3B-M, after epithelial damage there is loss of differentiated cells in $\Delta Np63$ depleted epithelium. However, the sentence “the proliferation of $\Delta Np63\Delta/\Delta; Rosa\Delta/\Delta$ tracheal epithelium at the 1-month time point post adenoviral cre injection (see Fig. 2j) is primarily due to basal cell proliferation and not a compensatory dedifferentiation of the ciliated or goblet cells.” is not clear.

The data previously shown in Fig. 2j (now relabelled as Fig. 2f), that is the increased number of Ki67 positive cells in tracheas of $\Delta Np63^{\Delta/\Delta}; Rosa^{\Delta/\Delta}$ vs. $\Delta Np63^{fl/fl}; Rosa^{M/M}$ mice at 1 month after infection with adenoviral cre, could be due to 2 distinct events: i) the increased proliferation of the basal cells; ii) the de-differentiation of other cell types (i.e. ciliated and goblet cells) from non-dividing cells into dividing cells. To distinguish between these 2 possibilities, we performed the experiment described in Fig. 3 by administering polidocanol (PDO), which causes the loss of all the suprabasal cells (e.g. ciliated and goblet cells). Because the basal cells (i.e. Krt5 positive cells) are the only ones left after PDO administration, the increased number of Ki67 positive cells observed in the $\Delta Np63^{\Delta/\Delta}; Rosa^{\Delta/\Delta}$ tracheas compared to their wildtype counterparts (Fig. 3j,k and z) must be in the basal cell population. This suggests that lack of $\Delta Np63$ makes basal cells more proliferative, and that the same – and not the de-differentiation of ciliated and goblet cells – may be responsible of what is observed in Fig. 2f.

6) For Figure 4A-D, authors should provide some representative images, even in Supplementary material. In the legend relative to figure 4E, authors should explain how do they mark the tracheosphere (EdU)? Same comment for figure 4T: what the red signal stands for?

As indicated in the methods, the EdU and Annexin V shown in Fig. 4a,d were performed with a highthroughput cell imaging system (Nexcelom Celigo), which does not allow for the acquisition of high-resolution images. However, we have acquired high-resolution images of the tracheospheres and the colonies of distal lung stem cells with an Olympus IX83 microscope. Both were imaged with the channel matching their respective *Rosa* allele: the

$\Delta Np63^{fl/fl}; Rosa^{M/M}$ cells express tdTomato and were imaged with the red channel, while the $\Delta Np63^{\Delta/\Delta}; Rosa^{\Delta/\Delta}$ cells express GFP and were imaged with the green channel. To avoid any confusion, the $\Delta Np63^{\Delta/\Delta}; Rosa^{\Delta/\Delta}$ tracheospheres shown in the previous version of Fig. 4e, that were previously pseudocolored in red, are now pseudocolored in green to match their respective endogenous fluorescence. Additionally, we now labelled the panels accordingly, as seen below in Figure 7, which is also included in the manuscript as new Fig. 4e and 4t.

Figure 7. a Representative wells of tracheosphere formation assay of $\Delta Np63^{fl/fl}; Rosa^{M/M}$ (tdTomato positive) and $\Delta Np63^{\Delta/\Delta}; Rosa^{\Delta/\Delta}$ (GFP positive) basal cells at passage 3. Scale bars equal 250 μm . b Representative images of colonies formed by $\Delta Np63^{fl/fl}; Rosa^{M/M}$ (tdTomato positive) and $\Delta Np63^{\Delta/\Delta}; Rosa^{\Delta/\Delta}$ (GFP positive) AT2 cells and BASCs. Scale bars equal 50 μm .

7) In figure 5G and 5H, it seems unrealistic that a $\Delta Np63$ motif would be 1-7kb long. Maybe the scale of the picture is wrong? Moreover, a different acetylation pattern between control cells and $\Delta Np63\Delta$ cells seems striking only for *KRT5* and *BCL9L* genes in tracheal cells (Fig. 5G, H), while in distal lung cells it is difficult to appreciate a difference in the acetylation patterns of *ETV5* and *BCL9L* genes with or without $\Delta Np63$. This needs to be mentioned and experimentally clarified.

We apologize for the confusion caused by our wording in describing the presence of the $\Delta Np63$ motif in these tracks. We have now clarified in the text that we are indicating which

among the H3K27ac peaks also contained the Δ Np63 motif shown in Fig. 5c and updated the tracks shown in Fig. 5g,h and Fig. 6d,e by labelling these regions as “H3K27ac peaks w/ Δ Np63 motif”. Regarding the difference in the acetylation patterns of *Etv5* and *Bcl9l* between the two genotypes, we have now added to the methods that the difference in the intensity of all the H3K27ac peaks considered in the manuscript was always significant ($q < 0.05$), and we included the scores for the top 2000 Δ Np63-regulated enhancers in basal cells and AT2 cells in the new Supplementary Tables 2 and 4, respectively. Additionally, mentioned in our reply to reviewer #1’s point #12, we have shown that Δ Np63 regulates the expression of *ETV5* and *BCL9L* both in mouse AT2 (see Fig. 6c) and human cancer cells (see Fig. 7c,d) by promoting the H3K27ac levels at the enhancers of these genes (see Fig. 7h,g), thus validating our ChIP-seq data.

8) *The authors need to analyze the presence of a positive correlation between Δ Np63 and BCL9L in LUAD and LUSC cohorts.*

Even though it was not possible for us to look for this correlation in LUAD and LUSC patients given the difficulty to discriminate between Δ Np63 and TAp63 in the TCGA datasets due to the insufficient number of reads in the isoform-specific exons of *TP63*, we tested for this correlation *in vitro*. As shown in Figure 6 above, which is now included in the manuscript as new Supplementary Fig. 3a,e, downregulation of *Δ Np63* in a panel of human LUAD and LUSC cell lines decreases the expression levels of *BCL9L*, thus providing additional data demonstrating that *BCL9L* is a novel Δ Np63 target gene.

9) *Rescue experiments in vitro showing that BCL9L overexpression in Δ Np63 cells may rescue oncogenic properties (stemness, anchorage-independent growth etc) are required.*

To address the reviewer’s request, we performed soft agar assays with 3 LUAD and 2 LUSC cell lines, where we downregulated Δ Np63 via siRNA and overexpressed *BCL9L*. As shown in Figure 8, which is also included in the manuscript as new Fig. 8c and new Supplementary Fig. 4a,c, the reduced anchorage-independent growth due to the reduced levels of Δ Np63 was

rescued by the overexpression of BCL9L, indicating that BCL9L is sufficient to mediate the oncogenic properties of Δ Np63 in both non-small cell lung cancer subtypes.

Figure 8. Quantification of colony formation efficiency in soft agar of the indicated cells transfected with the indicated constructs. Boxplots represent the individual data points, median and whiskers (min to max method) are shown, $n = 3$, * = $P < 0.05$, ** = $P < 0.01$, vs. respective siNT / Empty, two-tailed Student's t test.

Notably, when we performed a western blot analysis in the same cells used for the soft agar assays shown in Figure 8, we found that the protein levels of BCL9L decreased in all cell lines when Δ Np63 was downregulated, further supporting our data of the regulation of BCL9L expression by Δ Np63 (see Figure 9, which is also included in the manuscript as new Supplementary Fig. 4d,e).

Figure 9. Representative western blot analysis for the indicated proteins in the same LUAD (a) and LUSC (b) cell lines shown in Figure 8.

Reviewer #3 (Remarks to the Author):

Summary

In this manuscript, Napoli et al. dissect the role of DNp63 in NSCLC, including adenocarcinoma (LUAD) and squamous cell carcinoma (LUSC). The conditional allele that enables specific ablation of DNp63 in vivo and in vitro is a major strength of the study. Overall, the data associated with LUSC and its cell(s) of origin is convincing and either confirms or extends previous observations about the role of DNp63 in squamous cell carcinoma arising in different tissues. In particular, the analysis of DNp63 in tracheal basal cells is quite intriguing. However, the data on LUAD do not fully support the authors' conclusions and need to be expanded to adequately delineate the role of DNp63 in this disease.

We would like to thank the reviewer for deeming our data as intriguing and for providing constructive suggestions to improve our manuscript.

Major points

1) The data in Figure 1A clearly support a role for DNp63 at some stage of Kras-driven lung tumorigenesis. However, there are concerns with other panels in this figure as outlined below. Figure 1C. The IHC for DNp63 shown does not support the statement "Tumors from KrasG12D/+ mice stained positive for Δ Np63". The signal to noise is quite low in this image and it is difficult to believe that all tumor cells are positive. If a subset of cells are positive, it is hard to distinguish them. Additional characterization and precise description of DNp63 expression in control tumors is needed. Options could include: using a different antibody for IHC (such as the monoclonal antibody BC28, which has been used in multiple publications on human and mouse FFPE tissue); including positive control normal cells (such as basal cells) from the same slide to illustrate relative staining levels; and evaluation of recent single cell analyses of K and KP tumors (DOI: 10.1016/j.ccell.2020.06.012) to quantitate expression of DNp63 in those datasets. Since most human LUAD are thought to be DNp63-negative (with rare exceptions), characterization of DNp63 expression in this model needs to be very clear, particularly if the conclusion is that DNp63 expression in the mouse model of LUAD differs substantially from human LUAD.

We really appreciate the suggestion of using the BC28 antibody to detect Δ Np63 in Kras-driven LUAD via immunohistochemistry. We were able to more clearly show that Δ Np63 is expressed in these lesions and have now replaced the previous version of Fig. 1c with the new data obtained with the BC28 antibody and shown here as Figure 10.

Figure 10. Representative image of IHC for Δ Np63 in lung lesions from *Kras*^{G12D/+} mice. Scale bar equals 100 μ m.

2) Figures II/J-M, V: siRNA and shRNA experiments are difficult to interpret in the absence of information (immunoblotting) on baseline expression of Δ Np63 and the reduction in protein levels. (Specific nuclear positivity is difficult to identify in the IHC in IP/T.) The use of a single shRNA against Δ Np63 also makes the data less robust. At a minimum, it would be nice to know if this shRNA has no effect in cell lines lacking Δ Np63.

As requested by the reviewer, we have now included the data showing the downregulation of Δ Np63 by 2 independent sequences previously used by our laboratory (Napoli M. *et al.*, *Cancer Cell*. 2016 Jun 13;29(6):874-888. doi: 10.1016/j.ccell.2016.04.016. PMID: 27300436; and Bui N.H.B. *et al.*, *Cancer Res*. 2020 Jul 1;80(13):2833-2847. doi: 10.1158/0008-5472.CAN-19-2733. Epub 2020 Apr 20. PMID: 32312834). The most efficient of the 2 sequences (indicated in the manuscript as si Δ Np63 #1 and sh Δ Np63#1) is the one that was

utilized for the experiments with LUAD cells (see Fig. 1e) and LUSC cells (see Fig. 1f-r). The western blot analysis for $\Delta Np63$ (now included in the manuscript as new Supplementary Fig. 1b,c) is also shown above as Figure 2.

3) *Figures 2U-B' do not seem to be described in the manuscript.*

We thank the reviewer for pointing that out. We have now removed Fig. 2u,b' from the revised manuscript.

4) *Figure 4. Loss of DNp63 expression after Ad-Cre is not documented anywhere in the figure. This is particularly important for AT2/BASC studies, as these cells are not anticipated to have high levels of DNp63 at baseline. Relative levels of DNp63 in the different cell types are important to document.*

The validation of the loss of $\Delta Np63$ in AT2 cells is shown in Fig. 6c and demonstrates the efficient reduction of the $\Delta Np63$ expression after infection with adenoviral cre. We apologise if the qRT-PCR data shown in a separate figure compared to that of the AT2 colony culture data may have caused any confusion.

6) *“BCL9L is critical for maintenance of LUSC and LUAD” is a very broad claim to make when a single cell line of each tumor type has been evaluated. More cell lines and/or mining of public data (like DepMap) are needed. Were these experiments performed in the other LUAD cell lines from figure 1, all of which appear sensitive to DNp63 knockdown? Description of cell line genetics is warranted (how does driver mutation status compare with the GEMM?). As in figure 1, the use of a single shRNA and the absence of immunoblotting for target proteins is an issue. There is also no data directly demonstrating that BCL9L mediates the effects of DNp63 (rescue of shDNp63 by exogenous BCL9L, for example).*

As requested by the reviewer, we extended our analysis to a group of 3 LUAD and 2 LUSC cell lines, all of which express $Kras^{G12D/+}$ as now stated in both the results and the methods of the revised manuscript. As shown in Figure 8, which is also included in the manuscript as

new Fig. 8c and new Supplementary Fig. 4a,c, the overexpression of BCL9L is capable of rescuing the reduced growth in soft agar assays due to the reduced levels of $\Delta Np63$, thus indicating that BCL9L is an important oncogene mediating the oncogenic properties of $\Delta Np63$ in both LUAD and LUSC. Additionally, as mentioned in our reply to point #2, we have included the data showing the downregulation of $\Delta Np63$ with 2 independent sequences (see Figure 2 above, also included in the manuscript as new Supplementary Fig. 1b,c), the most effective of which was used in the 5 cell lines used for these rescue experiments and caused a reduction in *BCL9L* mRNA (see Figure 7, which is also included in the manuscript as new Fig. 8c and new Supplementary Fig. 4a,c) and protein levels (see Figure 9, which is also included in the manuscript as new Supplementary Fig. 4d,e).

Minor points

1) *“we assessed a greater than 90% recombination in the trachea and distal lung assessed by GFP expression from the ROSA allele”. This is not a very precise description...90% of all cells? Or of all epithelial cells? It should be described in more detail and documented if essential to the manuscript. Or it could potentially be removed.*

We have now included a representative image of the staining for GFP and tdTomato in the tracheal epithelium and distal lungs isolated from $\Delta Np63^{fl/fl};Kras^{LSL-G12D/+};Rosa^{M/M}$ and $\Delta Np63^{\Delta/\Delta};Kras^{G12D/+};Rosa^{\Delta/\Delta}$ mice collected at 20 weeks post-infection with adenoviral cre (see Figure 1 above, also included as new Supplementary Fig. 1a). As shown in the figure, our assessment of a greater than 90% recombination in both the trachea and the distal lungs of $\Delta Np63^{\Delta/\Delta};Kras^{G12D/+};Rosa^{\Delta/\Delta}$ mice is based on the GFP expression from the *Rosa* allele.

2) *Figure 1C. I have no doubt that there were tumors in the *Kras*G12D model, but the H&E image here contains a cluster of lymphocytes. A different image of a tumor should be taken. Figure 1D. The cells here appear to mainly include a junction between an airway and alveoli. Ideally, Figures 1C-F would compare tumors of the same grade (grade 1?) from each genotype so that there is an apples to apples comparison. Depicting AAH from each genotype would also be helpful in understanding which stage of tumorigenesis $\Delta Np63$ expression is most readily detectable.*

We have now replaced the previous panels with a representative image of the IHC for $\Delta Np63$ (BC28 antibody) in lung lesions from *Kras*^{G12D/+} mice (see Figure 10 above, which is also included in the manuscript as new Fig. 1c).

3) “...suggesting that $\Delta Np63$ serves to maintain the proliferation of distal lung stem cell populations (Fig. 1g,h).” This conclusion is not really supported by the observation.

We have now clarified in the text that the results are “*suggesting that $\Delta Np63$ is required for the maintenance of the distal lung stem cell populations*”.

4) Figure 5G/H – these seem to depict H3K27ac signal mainly at promoters – are there known enhancers for these genes, either in the depicted region or elsewhere.

Based on the reviewer’s suggestion, we have now compared the H3K27ac peaks that we identified in AT2 and basal cells with enhancers present in two distinct catalogues: Fantom 5 and the mouse Encode lung enhancer catalogues (Abugessaisa I. *et al.*, *Nucleic Acids Res.* 2021 Jan 8;49(D1):D892-D898. doi: 10.1093/nar/gkaa1054. PMID: 33211864; and Shen Y. *et al.*, *Nature.* 2012 Aug 2;488(7409):116-20. doi: 10.1038/nature11243. PMID: 22763441). Interestingly, this comparison allowed us to see an overlap of 17.02% and 16.46% between the known enhancers reported in either catalogue and the H3K27ac peaks in basal cells and AT2, respectively. This analysis is now added to the revised manuscript as new Supplementary Table 3. Additionally, we have also updated Fig. 5g,h and Fig. 6d,e to include the tracks for both Fantom 5 and the mouse Encode lung enhancer catalogues (see Fig. 11 in the next page).

5) Although it may be beyond the scope of a revision, if the authors have done any experiments with Cre driven by cell type-specific promoters (SPC-Cre, CCSP-Cre), this would be informative.

We agree with the reviewer that it would be informative to perform some of the experiments described in our manuscript with cell type-specific cre-drivers and they will be the focus of future studies.

Figure 11. ChIP-seq profiles for the indicated tracks in the *Krt5* (a) and *Bcl9l* (b) loci in basal cells of the indicated genotypes and in the *Etv5* (c) and *Bcl9l* (d) loci in basal cells of the indicated genotypes.

Reviewers' Comments:

Reviewer #1:

Remarks to the Author:

In the revised manuscript, the authors have addressed a number of my questions on the original data. However, there are still multiple concerns on the epigenomic analyses that need to be answered in order to improve the clarity and quality of the work.

1) There is a quite big revision now that the paper focuses on the top 2000 enhancers instead of super-enhancers. But some clarifications need to be provided: in fig. 5e and 6b, the legend reads "Ranking of top 2000 enhancers based on H3K27ac ChIP-seq". However, in the response letter and the manuscript, the authors mentioned that these are "the top 2000 Δ Np63-regulated enhancers" (for example in the response to #9). Which one is it?

2) In the response to question #12: "In the tracks shown in Fig. 5g,h and Fig. 6d,e we just wanted to highlight which among the H3K27ac peaks also contained this Δ Np63 motif. To clarify that, we have now corrected these figures by labelling these regions as "H3K27ac peaks w/ Δ Np63 motif". However, no P63 motif is presented in Fig. 6d. Plus, it is still very confusing and unusual to present a 5-kb long motif in Fig.5G, although I understand the authors' explanation.

3) In the response to question #7: " Δ Np63 regulated regions were identified by searching for the Δ Np63 motif in the H3K27ac peaks as shown in Fig. 5g,h and Fig. 6d,e." is confusing: so there were no peaks identified from the Δ Np63 ChIP-Seq? If so, the quality of the ChIP-Seq is questionable. But if there were peaks identified, why not use these peaks to define " Δ Np63 regulated regions". I disagree that " Δ Np63 regulated regions" can be defined by H3K27ac peaks containing Δ Np63 motif, since this would generate numerous false positive calling.

4) Regarding the question #8, the revised Fig.5b is still showing "p value" instead of "FDR-adjusted p-value". It needs to be updated.

5) In the top500 enhancers, regardless of the p63 motif, they are all significantly decreased in the KO cells, indicating that many of these changes are likely indirect effect, or even batch effect. This is even more so in Fig.6b, where all top2000 enhancers are considerably reduced, which cannot be explained by the direct effect of p63 regulation. How many replicates were used for the ChIP-Seq? And the robustness of the correlation between the replicates?

6) In line 294 "Genes associated with the top 2,000 enhancers regulated by Δ Np63 are involved in epithelialization and cell junction maintenance (Fig. 5e,f).".

How was this conclusion derived? Fig. 5e,f only highlighted a few genes. However, based on the new Fig.7a, these 2000 enhancers are associated with almost 8,000 genes. A more global analysis is needed for this conclusion. Otherwise, the authors should revisit this statement.

Reviewer #2:

Remarks to the Author:

The revised version of the manuscript is compelling with the specific comments raised previously by this reviewer. The manuscript is now suitable for full acceptance in Nat. Comm.

Reviewer #3:

Remarks to the Author:

I am satisfied with the response of the authors to most of my concerns. However, I do not think this statement is correct:

"Tumors from KrasG12D/+ mice stained positively for Δ Np63 (Fig. 1c), indicating that Δ Np63 is indeed robustly expressed in Kras-driven LUAD."

In my own lab's experience with the K and KP lung models, we rarely if ever see robust expression of DNp63 in the predominant lung tumors from these mice, specifically those that are found in the alveolar space. The new picture provided by the authors appears to be a hyperplasia arising in an airway (Fig 1c). These hyperplasias are commonly found when using CMV-Cre to activate Kras, but are distinct from the main tumors that grow predominantly in the alveolar space. Although the DNp63 staining in this hyperplasia looks real/specific, I would be surprised if all the alveolar tumors in the K model have robust DNp63 expression, particularly when compared to internal positive controls that express DNp63 (such as basal cells in the upper airways/trachea).

I think the authors' data suggest that DNp63 is very important for an early (undefined) stage of tumorigenesis in the Kras model (based on figs 1a-b), despite the fact that its protein levels are fairly low relative to other normal cell types that express DNp63 in vivo (like basal cells). Although counterintuitive, this is still an important result.

I think it would not be good for the field to publish the paper without a thorough and accurate analysis of DNp63 expression in the Kras LUAD model (ideally with carefully titrated antibody and review of the slides by a pathologist), and I don't think figure 1C meets that standard.

Response to Reviewers' Comments:

We would like to thank again the reviewers for their constructive suggestions. We have addressed all the points raised by the reviewers and believe that the manuscript is of high significance to the cancer stem cell and super-enhancer fields. We now provide additional evidence of the expression of $\Delta Np63$ in Kras-driven lung adenocarcinomas. Additionally, we have now included the $\Delta Np63$ ChIP-seq signals for $\Delta Np63^{fl/fl}; Rosa^{M/M}$ basal cells in the *Krt5* and *Bcl9l* loci, and the $\Delta Np63$ ChIP-seq signals for $\Delta Np63^{fl/fl}; Rosa^{M/M}$ AT2 cells in the *Etv5* and *Bcl9l* loci. Finally, the RNA-seq of the $\Delta Np63^{fl/fl}; Rosa^{M/M}$ and $\Delta Np63^{A/A}; Rosa^{A/A}$ basal cells, as well as the ChIP-seq of the $\Delta Np63^{fl/fl}; Rosa^{M/M}$ and $\Delta Np63^{A/A}; Rosa^{A/A}$ basal and AT2 cells have been deposited to NCBI Gene Expression Omnibus (GEO) repository (series GSE131671, token = qbqpysoqnpsdpyh). Our specific point-by-point response to each comment is below in boldface type.

Reviewers' comments:

Reviewer #1 (Remarks to the Author):

In the revised manuscript, the authors have addressed a number of my questions on the original data. However, there are still multiple concerns on the epigenomic analyses that need to be answered in order to improve the clarity and quality of the work.

1) There is a quite big revision now that the paper focuses on the top 2000 enhancers instead of super-enhancers. But some clarifications need to be provided: in fig. 5e and 6b, the legend reads "Ranking of top 2000 enhancers based on H3K27ac ChIP-seq". However, in the response letter and the manuscript, the authors mentioned that these are "the top 2000 $\Delta Np63$ -regulated enhancers" (for example in the response to #9). Which one is it?

These two figures show the ranking of the top 2,000 enhancers based on H3K27ac ChIP-seq in $\Delta Np63^{fl/fl}; Rosa^{M/M}$ and $\Delta Np63^{A/A}; Rosa^{A/A}$ basal (Fig. 5e) and AT2 (Fig. 6b) cells. We have now revised the manuscript to clarify that.

2) In the response to question #12: “In the tracks shown in Fig. 5g,h and Fig. 6d,e we just wanted to highlight which among the H3K27ac peaks also contained this Δ Np63 motif. To clarify that, we have now corrected these figures by labelling these regions as “H3K27ac peaks w/ Δ Np63 motif”. However, no P63 motif is presented in Fig. 6d. Plus, it is still very confusing and unusual to present a 5-kb long motif in Fig. 5G, although I understand the authors’ explanation.

To avoid any further confusion and as also suggested by the reviewer in point #3 below, we have now removed the “H3K27ac peaks w/ Δ Np63 motif” tracks from Fig. 5g,h and Fig. 6d,e, and replaced them with the tracks of the Δ Np63 ChIP-seq signal showing the recruitment of Δ Np63 to these 4 loci in Δ Np63^{fl/fl};Rosa^{M/M} basal and AT2 cells (see Figure 1 below, also included in the manuscript as updated Fig. 5g,h and Fig. 6e,f). Additionally, we are now indicating in green the regions homologous to the human chromatin locations validated for the recruitment of Δ Np63 in the model utilizing dCas9 technology (Fig. 7c-e) and in lung cancer cells (Fig. 7f-h).

Figure 1. ChIP-seq profiles for $\Delta Np63^{fl/fl}; Rosa^{M/M}$ basal cells in the *Krt5* (a) and *Bcl9l* (b) loci and for $\Delta Np63^{fl/fl}; Rosa^{M/M}$ AT2 cells in the *Etv5* (c) and *Bcl9l* loci (d).

3) In the response to question #7: “ $\Delta Np63$ regulated regions were identified by searching for the $\Delta Np63$ motif in the H3K27ac peaks as shown in Fig. 5g,h and Fig. 6d,e.” is confusing: so there were no peaks identified from the $\Delta Np63$ ChIP-Seq? If so, the quality of the ChIP-Seq is questionable. But if there were peaks identified, why not use these peaks to define “ $\Delta Np63$

regulated regions”. I disagree that “ Δ Np63 regulated regions” can be defined by H3K27ac peaks containing Δ Np63 motif, since this would generate numerous false positive calling.

As stated in our reply to point #2, we followed the reviewer’s suggestion and replaced the “H3K27ac peaks w/ Δ Np63 motif” tracks with the tracks of the Δ Np63 ChIP-seq signal. These data shown above in Figure 1 and also included in the manuscript as updated Fig. 5g,h and Fig. 6e,f indicate that Δ Np63 binds to these chromatin regions in Δ Np63^{fl/fl};Rosa^{M/M} basal and AT2 cells, similarly to what is shown in two distinct human cancer cell lines (Fig. 7c-h).

4) Regarding the question #8, the revised Fig.5b is still showing “p value” instead of “FDR-adjusted p-value”. It needs to be updated.

We thank the reviewer for pointing that out. We have now corrected the x-axis of Fig. 5b accordingly.

5) In the top500 enhancers, regardless of the p63 motif, they are all significantly decreased in the KO cells, indicating that many of these changes are likely indirect effect, or even batch effect. This is even more so in Fig.6b, where all top2000 enhancers are considerably reduced, which cannot be explained by the direct effect of p63 regulation. How many replicates were used for the ChIP-Seq? And the robustness of the correlation between the replicates?

The ChIP-seq was performed by pooling either primary basal or AT2 cells collected from at least 5 Δ Np63^{fl/fl};Rosa^{M/M} mice, thus compensating for any possible variability across mice and guaranteeing for the robustness of the signals. All the observed effects, including those shown in Fig.6b (now listed as Fig. 6c), are associated to Δ Np63. Even though we cannot rule out all indirect effects due to the loss of Δ Np63, the targets validated via qRT-PCR (Fig. 5f, Fig. 6d, and Fig. 7c-e) and those validated via ChIP assays (Fig. 7f-h) are directly regulated by Δ Np63. The ChIP-seq data for both the basal and the AT2 cells have been deposited to NCBI Gene Expression Omnibus (GEO) repository (series GSE131671), and can be accessed by using the following token: qbqpysoqnpsdpyh.

6) In line 294 “Genes associated with the top 2,000 enhancers regulated by Δ Np63 are involved in epithelialization and cell junction maintenance (Fig. 5e,f).”.

How was this conclusion derived? Fig. 5e,f only highlighted a few genes. However, based on the new Fig.7a, these 2000 enhancers are associated with almost 8,000 genes. A more global analysis is needed for this conclusion. Otherwise, the authors should revisit this statement.

We have now revised the sentence as suggested by the reviewer to clarify that “among the genes associated with the top 2,000 enhancers in basal cells, there are genes involved in epithelialization and cell junction maintenance”.

Reviewer #2 (Remarks to the Author):

The revised version of the manuscript is compelling with the specific comments raised previously by this reviewer. The manuscript is now suitable for full acceptance in Nat. Comm.

We would like to thank the reviewer for deeming our manuscript suitable for full acceptance in *Nature Communications*.

Reviewer #3 (Remarks to the Author):

I am satisfied with the response of the authors to most of my concerns. However, I do not think this statement is correct:

"Tumors from $Kras^{G12D/+}$ mice stained positively for $\Delta Np63$ (Fig. 1c), indicating that $\Delta Np63$ is indeed robustly expressed in $Kras$ -driven LUAD."

In my own lab's experience with the K and KP lung models, we rarely if ever see robust expression of $\Delta Np63$ in the predominant lung tumors from these mice, specifically those that are found in the alveolar space. The new picture provided by the authors appears to be a hyperplasia arising in an airway (Fig 1c). These hyperplasias are commonly found when using CMV-Cre to activate $Kras$, but are distinct from the main tumors that grow predominantly in the alveolar space. Although the $\Delta Np63$ staining in this hyperplasia looks real/specific, I would be surprised if all the alveolar tumors in the K model have robust $\Delta Np63$ expression, particularly when compared to internal positive controls that express $\Delta Np63$ (such as basal cells in the upper airways/trachea).

I think the authors' data suggest that $\Delta Np63$ is very important for an early (undefined) stage of tumorigenesis in the $Kras$ model (based on figs 1a-b), despite the fact that its protein levels are fairly low relative to other normal cell types that express $\Delta Np63$ in vivo (like basal cells). Although counterintuitive, this is still an important result.

I think it would not be good for the field to publish the paper without a thorough and accurate analysis of $\Delta Np63$ expression in the $Kras$ LUAD model (ideally with carefully titrated antibody and review of the slides by a pathologist), and I don't think figure 1C meets that standard.

Based on the reviewer's suggestion, we have now included in the revised manuscript a panel of 8 representative images of IHC for $\Delta Np63$ in lung lesions from 3 $Kras^{G12D/+}$ mice. As seen in these images shown below in Figure 2 and quantified in Figure 3, also included in the manuscript as new Supplementary Fig. 1b,i and new Fig. 1c, respectively, $\Delta Np63$ is expressed in cancer cells of $Kras^{G12D/+}$ lungs in a heterogenous manner. To reflect that, we have now revised the text to state that "Tumours from $Kras^{G12D/+}$ mice stained positively yet heterogeneously for $\Delta Np63$ (Fig. 1c and Supplementary Fig. 1b,i), indicating that $\Delta Np63$ is indeed expressed in $Kras$ -driven LUAD".

Figure 2. Representative images of IHC for Δ Np63 in lung lesions from *Kras*^{G12D/+} mice. Scale bars equal 800 μ m (bottom left corners) and 200 μ m (bottom right corners). Dashed lines outline the areas where the percentage of Δ Np63 positive cells was quantified.

Figure 3. Quantification of the percentage of $\Delta Np63$ positive cells of the tumours shown in Figure 2.

Reviewers' Comments:

Reviewer #1:

Remarks to the Author:

In this 2nd revision, the authors have addressed my remaining concerns. The quality of the computational analyses and epigenomic data have been improved.

Reviewer #3:

Remarks to the Author:

I am satisfied with the authors' response to my most recent set of comments. The article is ready for publication.

Response to Reviewers' Comments:

We would like to thank once again all the reviewers for their suggestions and comments throughout the revision process that helped us in improving our manuscript. Our specific point-by-point response to each of their comments is below in boldface type.

Reviewers' comments:

Reviewer #1 (Remarks to the Author):

In this 2nd revision, the authors have addressed my remaining concerns. The quality of the computational analyses and epigenomic data have been improved.

We thank the reviewer for his/her suggestions and for considering the quality of our data improved.

Reviewer #3 (Remarks to the Author):

I am satisfied with the authors' response to my most recent set of comments. The article is ready for publication.

We thank the reviewer for his/her comments and for deeming our response to be exhaustive.